# The potential of synthetic aperture radar interferometry for assessing meltwater lake dynamics on Antarctic ice shelves

Weiran Li [1], Stef Lhermitte [1], and Paco López-Dekker [1]

[1]Department of Geoscience and Remote Sensing, Delft University of Technology, Delft, The Netherlands

**Correspondence:** Weiran Li (w.li-7@tudelft.nl)

**Abstract.** Surface meltwater drains on several Antarctic ice shelves, resulting in surface and sub-surface lakes that are potentially critical for the ice shelf collapse. Despite these phenomena, our understanding and assessment of the drainage and refreezing of these lakes is limited, mainly due to lack of field observations and to the limitations of optical satellite imagery during polar night and in cloudy conditions. This paper explores the potential of backscatter intensity and of interferometric coherence and phase from synthetic aperture radar (SAR) imagery as an alternative to assess the dynamics of meltwater lakes. In four case study regions over Amery and Roi Baudouin ice shelves, East Antarctica, we examine spatial and temporal variations in SAR backscatter intensity and interferometric (InSAR) coherence and phase over several lakes derived from Sentinel-1A/B C-band SAR imagery. Throughout the year, the lakes are observed in completely frozen state, in partially frozen state with a floating ice lid, and as open water lakes. Our analysis reveals that the meltwater lake delineation is challenging during the melting period when the contrast between melting snow and lakes is indistinguishable. Despite this finding, we show using a combination of backscatter and InSAR observations that lake dynamics can be effectively captured during other non-summertime months. Moreover, our findings highlight the utility of InSAR-based observations for discriminating between refrozen ice and subsurface meltwater, and indicate the potential for phase-based detection and monitoring of rapid meltwater drainage events. The potential of this technique to monitor these meltwater change events is, however, strongly determined by the satellite revisit interval and potential changes in scattering properties due to snowfall or melt events.

## 1  Introduction

Widespread surface meltwater has been observed on Antarctic ice shelves over the past century (Kingslake et al., 2017). Through seasonal formation and draining of supraglacial lakes, which have the potential to fracture and weaken ice shelves through repeated compression and uplift, respectively (Banwell et al., 2013), such phenomena may have important implications for ice-shelf hydrofracture and collapse (Bell et al., 2018). Therefore, accurately observing the spatial and temporal evolution (filling, drainage or refreezing) of such lakes is pertinent to elucidating the future stability and response of the Antarctic Ice Sheet to climate change.

Given the remote location, widespread area, and harsh climatic conditions in which these lakes form, satellite remote sensing has become the primary method of observing their evolution and dynamics (Brucker et al., 2010; Dirscherl et al., 2021). Previous studies exploited various satellite remote sensing data sources to observe these phenomena; for example, Kingslake

et al. (2017) presented an overview of the Antarctic-wide meltwater hydrological network by combining Landsat, WorldView and Aster optical satellite imagery together with historic (pre-satellite) aerial photography. Other work has combined both optical and synthetic aperture radar (hereafter SAR) imagery to detect meltwater features in both Greenland and Antarctica (Benedek and Willis, 2021; Dirscherl et al., 2021), including the detection of subsurface meltwater across East Antarctica's

Roi Baudouin Ice Shelf (RBIS; Lenaerts et al. (2016)). Such subsurface melting is not detectable from optical-based imagery alone (Miles et al., 2017), emphasising the potential utility of SAR to better detect total surface meltwater presence.

Despite the potential of optical imagery and SAR imagery in observing surface meltwater, both sensor types have limitations over Antarctica. Polar nights and cloud cover, for example, limit data coverage in optical-based imagery (Williamson et al., 2017), whereas the operating frequencies and active-source configuration of SAR sensors allow for all-weather, day-night

imaging (Miles et al., 2017). Relative to the intuitive representation of meltwater features detected by optical sensors, however, the interpretation of SAR imagery can be complex due to ambiguous backscatter returns and/or image geometry effects (e.g. Fahnestock et al. (1993); Miles et al. (2017); Rizzoli et al. (2017)). While cross-polarised (HV or VH) backscatter intensity SAR images generally provide a better contrast between water and ice than single polarisation (e.g. HH) images (Miles et al., 2017), such images are not necessarily always available over Antarctica (Hillebrand et al., 2021).

A potential solution to these limitations is interferometric processing of the synthetic aperture radar data (InSAR), which provides complementary information on the geometric and dielectric properties of the meltwater features. InSAR processing uses pairs of images of the same area separated by a particular temporal baseline to derive coherence and interferometric phase information. Coherence is an indicator of changes in the relative position of the scatterers between the two acquisitions, whereas the interferometric phase measures their range difference from the satellites with the precision of a fractional component of the

measuring radar wavelength. For high coherence areas, the phase can be related to a line-of-sight displacement without change in scattering properties (e.g. without intense regional precipitation and melts), whereas for low coherence areas where surface melts typically occur, the phase becomes scarcely informative (Hanssen, 2001). We expect this combination of coherence and phase information from InSAR to facilitate the continuous monitoring of meltwater dynamics. The changes in InSAR coherence have been proven useful in X-band for monitoring the refreeze of thermokarst lakes in the Arctic region (Antonova

et al., 2016). So far, however, no analysis has been conducted for C-band time series. Additionally, the interferometric phase might reveal information about the drainage and filling of lakes, as these processes result in a vertical displacement of the surface (Banwell et al., 2013). However, the value added using InSAR for such applications has not yet been examined.

In this paper, we assess the potential of C-band InSAR data to quantify the dynamic behaviour of meltwater filling, drainage and refreezing. For this purpose, we use a combination of backscatter, coherence and phase information to monitor recent

meltwater features over two East Antarctic locations—the Amery and Roi Baudouin (RBIS) ice shelves—using data collected by Sentinel-1A/B in 2017/2018. To supplement the interpretation of our (In)SAR-based analyses, we also utilise spatially and temporally collocated optical and passive microwave satellite data and climate data.

## 2 Data & Methods

### 2.1 Study areas

Two ice shelves in East Antarctica with well-known meltwater dynamics (Kingslake et al., 2017) are used as case studies. The first case study is on the Roi Baudouin Ice Shelf (RBIS), where in situ research was conducted and the exact locations of several lakes were mapped during field campaigns (Lenaerts et al., 2016; Dunmire et al., 2020). We use the supraglacial and englacial lakes mapped by Lenaerts et al. (2016) as delineated meltwater lake features, and complement that data set with manually delineated sample polygons of snow and ice surfaces based on Landsat imagery for studying the difference between meltwater lakes and the solid surrounding regions (Fig. 1).

For the second case study over Amery ice shelf, we use a similar approach based on sampled lake, snow and ice regions. For Amery, no previously published dataset from in situ studies is available. Therefore, samples of lakes are mapped manually based on available Landsat 8 imagery (introduced in Section 2.2) in summer 2017-2018. The goal of this sampling is not to map all possible lakes, but to get a representative sample polygon for each snow/ice/lake class.

### 2.2 Data

Two types of Level-1 Sentinel-1 Interferometric Wide (IW) products are used in this study: Single Look Complex (SLC) products, consisting of complex-valued data that preserve the phase information of the returned echoes, and Ground Range Detected (GRD) products, consisting of multi-looked backscatter intensity without phase information. For both products, HH-polarisation is used as this is the only polarisation widely available over the studied ice shelves.

Sentinel-1 SLC data are available on Copernicus Open Access Hub (Copernicus, 2014) and are processed to derive phase information and $\sigma^0$. SLC processing is carried out using the Delft Object-oriented Radar Interferometric Software (DORIS, http://doris.tudelft.nl), whose processing chain is summarised in Fig. 2. The co-registration between images is performed using magnitude images of the complex data. Sentinel-1 IW operates in Terrain Observation by Progressive Scans (TOPS, De Zan and Monti Guarnieri (2006)) mode, therefore phase ramps are accounted for via deramp and reramp processes to ensure co-registration accuracy (Yague-Martinez et al., 2016). For the retrieval of the sub-pixel azimuth shift, Enhanced Spectral Diversity (ESD) is also applied (Prats-Iraola et al., 2012; Yague-Martinez et al., 2017). Georeferencing is based on TanDEM-X digital elevation model (DEM) for RBIS (Lenaerts et al., 2016) and WGS84 geoid for Amery as it is the default DEM input of DORIS when TanDEM-X DEM of the same quality is not available at the time of processing. The final SLC products have an azimuth resolution of 20 m and a ground range resolution of 5 m (Torres et al., 2012).

GRD products are used mainly as supplementary backscatter intensity information when specific SLC tracks are not available (i.e. data from ascending track 59 before July 2017 are not available on the Copernicus Open Access Hub, as shown in Table 1). The GRD data are primarily acquired from Google Earth Engine (GEE), whose processing includes thermal noise removal, radiometric calibration, and terrain correction. The final backscatter product has a 10 m×10 m resolution. When normalised by the area of the resolution cell on the ground, the calibrated backscatter intensities are usually termed sigma-nought ($\sigma^0$), and this is the term we will use for the remaining of the paper for backscatter intensity.

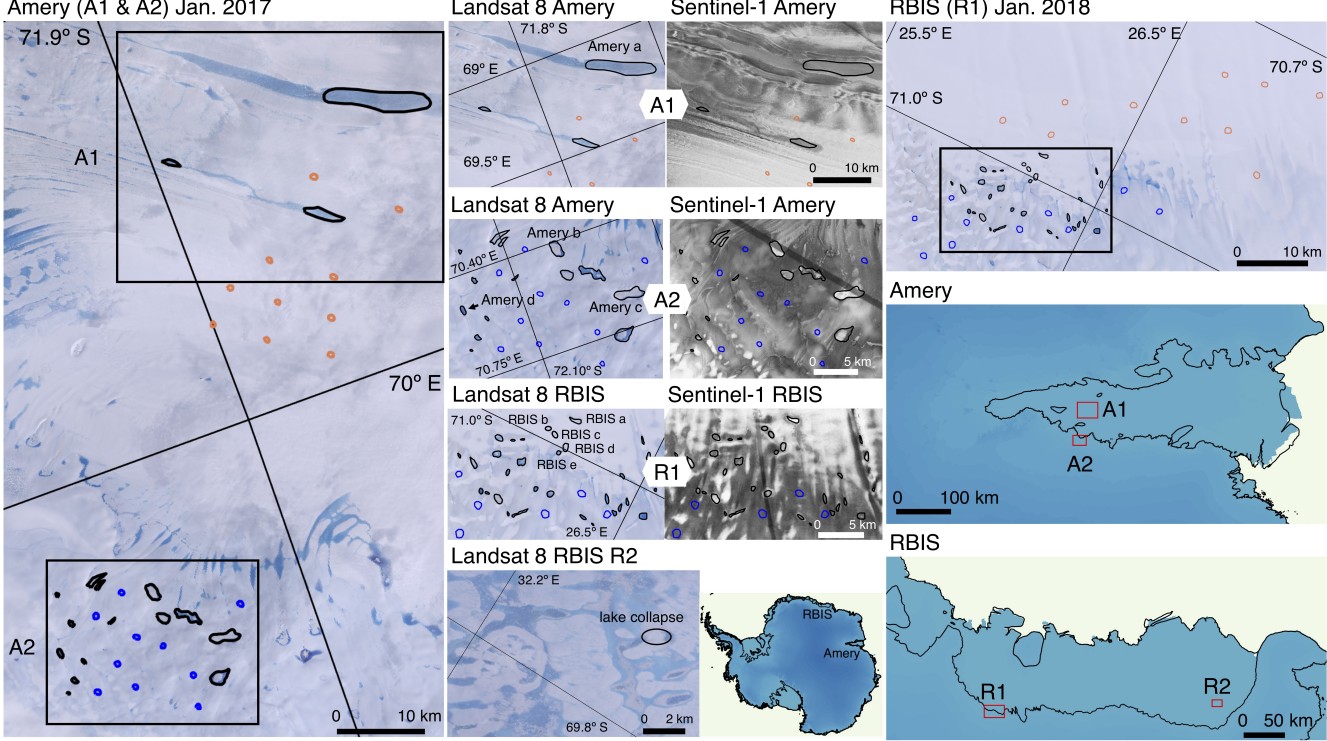

**Figure 1.** Outline of the Amery and Roi Baudouin Ice Shelf (RBIS) study areas (referred to as A1, A2, R1, and R2). Details of the investigated meltwater features are shown in both Landsat 8 true colour images and Sentinel-1 backscatter intensities. The images are acquired from Google Earth Engine (Gorelick et al., 2017). In all panels, the lakes used for the temporal backscatter and coherence analysis are delineated as black curves. The labels of the lakes correspond to the time series in Fig. 3. Snow (in orange) and ice (in blue) are also delineated for comparison against backscatter intensity and coherence values observed over lakes (Fig. 3). Panel R2 illustrates the lake feature shown in Fig. 9. The analysed ice shelves are highlighted in the Antarctica map, and the specific locations of A1, A2, R1 and R2 are shown in the Amery and RBIS maps. The DEM used as the background is from the REMA project (Howat et al., 2019), courtesy of the Polar Geospatial Center. The coastline is from the SCAR Antarctic Digital Database (Gerrish et al., 2021).

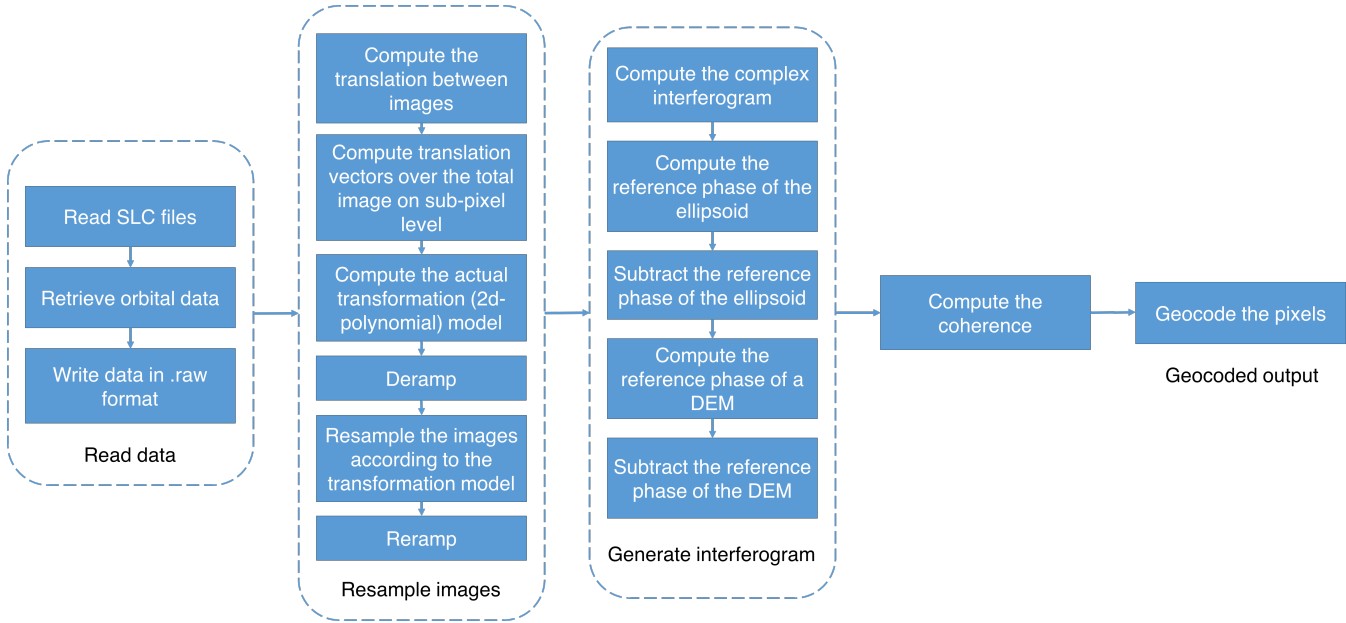

**Figure 2.** Delft Object-oriented Radar Interferometric Software (DORIS) processing flowchart from the software documentation available online at http://doris.tudelft.nl/software/doris_v4.02.pdf, customised based on Nikaein et al. (2021).

Additionally, independent datasets are used to help interpret the Sentinel-1 SAR data. First, Landsat 8 images are used for visual interpretation, i.e. solid snow and ice surfaces are shown in the images in white, and ice and lakes as a result of intensive melt are shown in blue. Available calibrated top-of-atmosphere (TOA) Tier 1 Landsat surface reflectance data (Chander et al., 2009) of true colour bands (bands 4, 3, and 2) and panchromatic band (band 8) are acquired from GEE at their native pixel resolution (30 m for RGB bands and 15 m for panchromatic band) without any additional pre-processing steps. Detailed data type and acquisition dates of satellite imagery are provided in Table 1.

To interpret temporal variations of Sentinel-1 backscatter intensity and coherence, it is moreover important to understand temporal melt extent and precipitation, as these are the potential drivers of changes in scatterers. For estimating melt extent, multi-frequency radiometer observations, more specifically, brightness temperature ($Tb$) measurements from the Special Sensor Microwave Imager/Sounder (SSMIS) sensors (Kunkee et al., 2008) are used. Brightness temperatures in polar stereographic projection are available on National Snow and Ice Data Center (NSIDC) (Meier et al., 2021).

Precipitation from ERA5 Daily Aggregates (Copernicus Climate Change Service (C3S), 2017) over A2 and R1 (in Fig. 1) in 5 km resolution is averaged spatially and acquired from GEE. Acquisition dates of the brightness temperature observations and ERA5 data overlap with the SLC acquisition dates from ascending track 59 and descending track 3 in Table 1.

**Table 1.** List of the imagery used in this study. When the end date is not specified, the table entry refers to a single acquisition. For SLC data from descending track 3, the repeat cycle is mainly 6 days, except that between Jan. 4 and Jan. 16, 2017 the revisit time is 12 days, and there is a lack of data on May 16, Sep. 13 and Sep. 19, 2017. For SLC data from ascending track 59, there is a lack of data on Feb. 26, 2018 and Mar. 10, 2018.

| Ice shelf (region) | Product | Track No. | Starting date | End date | Repeat cycle |
|---|---|---|---|---|---|
| RBIS (R1) | Sentinel-1 IW SLC | Ascending 59 | 2017/07/25 | 2018/04/15 | 12 days |
| RBIS (R2) | Sentinel-1 IW SLC | Descending 136 | 2017/12/04 | 2018/04/15 | 12 days |
| RBIS (R1) | Sentinel-1 IW GRD | Multiple | 2016/06/01 | 2018/05/31 | N/A |
| RBIS (R1) | Landsat 8 | Path 157 Row 110 | 2017/09/26 | - | N/A |
| RBIS (R2) | Landsat 8 | Path 155 Row 110 | 2017/12/01 | - | N/A |
| RBIS (R2) | Landsat 8 | Path 153 Row 110 | 2017/12/19 | - | N/A |
| RBIS (R1) | Landsat 8 | Path 156 Row 110 | 2018/01/09 | - | N/A |
| Amery (A1, A2) | Sentinel-1 IW SLC | Descending 3 | 2017/01/04 | 2018/01/17 | 12 or 6 days |
| Amery (A2) | Sentinel-1 IW GRD | Multiple | 2016/06/01 | 2018/05/31 | N/A |
| Amery (A1) | Landsat 8 | Path 127 Row 111 | 2017/01/27 | - | N/A |
| Amery (A2) | Landsat 8 | Path 126 Row 111 | 2017/10/03 | - | N/A |
| Amery (A1, A2) | Landsat 8 | Path 127 Row 111 | 2018/01/14 | - | N/A |

## 2.3 Methods

To assess meltwater lake dynamics, we analyse the spatial and the temporal variations of Sentinel-1 backscatter intensity and coherence over the lakes and control (snow/ice) sites. Therefore, we compare the spatial and temporal characteristics of the identified lakes with their surroundings to assess how well they can be distinguished in different seasons. For this purpose, the temporal variations in $\sigma^0$ and coherence are compared per lake, snow, ice class by analysing their time series of the mean and standard deviation for each class (i.e. lakes, snow and ice). In this comparison, 10 samples of snow and 10 samples of ice on each ice shelf are used as shown in Fig. 1. Second, the spatio-temporal variation in $\sigma^0$ is analysed along cross-sectional transects across the largest lake dimension to assess the seasonal differences between the lakes and their surrounding areas. This is a biennial analysis, in order to show that the lakes may not behave identically every year. Subsequently, individual images are analysed, where changes in $\sigma^0$ are compared to changes in coherence and phase to assess the added value of combining SAR backscatter intensity with InSAR information to improve the understanding of the melt-refreeze process of lakes.

Time series of backscatter intensity and coherence are interpreted with the assistance of melt extent and precipitation time series. As an approximation of melt extent, the Cross Polarisation Gradient Ratio (XPGR) meltwater detection method proposed by Abdalati and Steffen (1995) is applied, where horizontally polarised 19 GHz (19H) and vertically polarised 37 GHz (37V) brightness temperatures are used to calculate the XPGR:

$$XPGR = \frac{Tb_{19H} - Tb_{37V}}{Tb_{19H} + Tb_{37V}} \tag{1}$$

When XPGR ratio exceeds a specific threshold, the surface is assumed to experience melting. For SSMIS, this threshold is set as -0.0158 (Johnson et al., 2020). 19H and 37V observations used for the computation are measured daily and provided in 25 km resolution. In addition, time series of precipitation from ERA5 acquired from GEE are used directly.

## 3  Results

### 3.1  Backscatter intensity analysis

The mean $\sigma^0$ time series of lakes, snow and ice (Section 2.2) display strong seasonal variability, consistent with the changing nature of both surface snow and ice properties and the evolution of supraglacial lakes through time (Fig. 3). On Amery Ice Shelf, our observations reveal that $\sigma^0$ has different values for snow ($\sim$0 dB), lakes ($\sim$-5 dB) and ice ($\sim$-10 dB) and is relatively constant during the observed time span (fluctuations within $\sim$1 dB), with the exception of the summer melt seasons (January and February). In summer seasons, as a result of melting, the $\sigma^0$ of (wet) snow and lakes shows a strong decrease due to the change in dielectric constant. The $\sigma^0$ time series on RBIS show a similar pattern (i.e., $\sigma^0_{snow} > \sigma^0_{lake} > \sigma^0_{ice}$) except for Dec. 2017 and Jan. 2018, where $\sigma^0_{snow}$ drops below $\sigma^0_{lake}$ and $\sigma^0_{ice}$. Both the Amery and RBIS time series show, however, that the discrimination of lakes based on $\sigma^0$ alone is not straightforward as the $\sigma^0$ of the lakes often resembles the $\sigma^0$ of snow and ice.

A similar confusion between lakes and snow/ice samples is visible in the spatio-temporal analysis of selected cross-sectional transects. In the case of both RBIS a and Amery d (location shown in Fig. 1), for example, backscatter time series show significant inter-annual variation (Fig. 4). For RBIS a, this starts with high $\sigma^0$ values (similar to snow) with limited spatial variation in June–Nov. 2016, followed by a strong area-wide decrease in $\sigma^0$ during the melting season (Dec. 2016–Feb. 2017). Subsequently, a clear spatial pattern emerges with borders of low $\sigma^0$ at the edges and high $\sigma^0$ in the central regions, which respectively refer to the edge and central regions of the lake. This pattern is followed again by a new area-wide decrease in $\sigma^0$ in the Dec. 2017–Jan. 2018 melting season. This development is consistent with the description of ice lids in (Antonova et al., 2016) and the potential development of ice lids in winter on RBIS (Dunmire et al., 2020).

For Amery d, these spatio-temporal transect patterns of the lake are less distinguishable from the surrounding ice area, as the $\sigma^0$ of the lake closely resembles the $\sigma^0$ of the surrounding ice, except for Mar.–May 2018 when it shows a strong increase.

### 3.2  Coherence analysis

The coherence time series show a completely different behaviour than the $\sigma^0$ time series (Fig. 3). On Amery Ice Shelf, for example, snow, ice and lakes all have low or null coherence in summer, because of the altering scattering properties due to meltwater content. For the ice and snow zones, the coherence rises abruptly when the surface refreezes in spring, while the coherence over the lakes rises only gradually until winter, when the lakes reach coherence values that are similar to snow and ice. During winter, the coherence values of snow, ice and lakes show a similar behaviour with large temporal variations which fluctuate between 0.2 and 0.6 between successive (6-day) image acquisitions. These sudden drops likely result from short-term, weather-induced changes in scattering properties, including snowfall events (Fig. 5a). These drops are however sparse as the

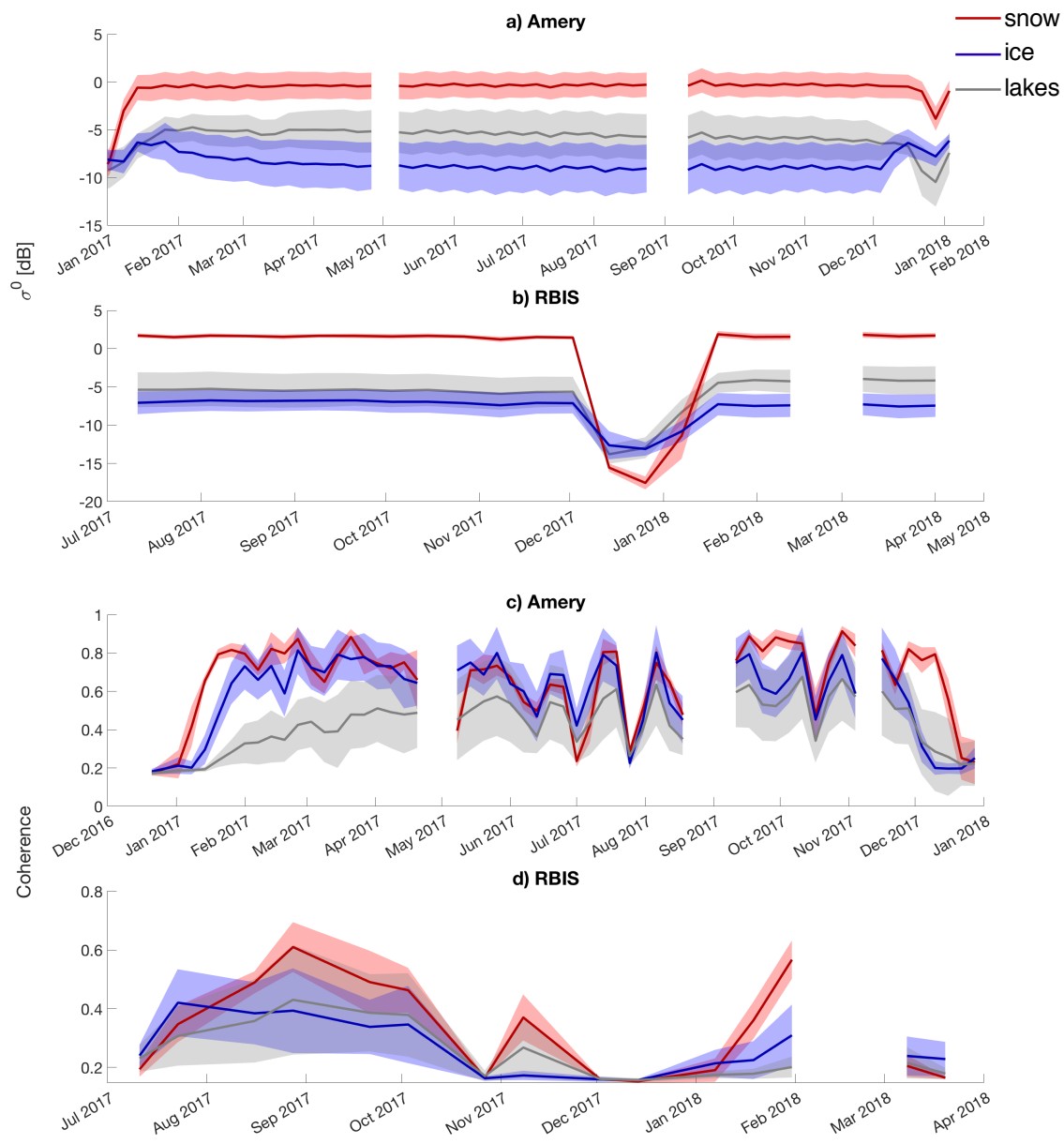

**Figure 3.** Time series of mean (solid line) and standard deviation (semi-transparent area) of $\sigma^0$ and coherence over Amery and Roi Baudouin ice shelves (see Fig. 1 for locations). Mean and standard deviation are calculated from all features indicated in Fig. 1. Times with a lack of 6/12-day revisit frequency are masked, resulting in discontinuities.

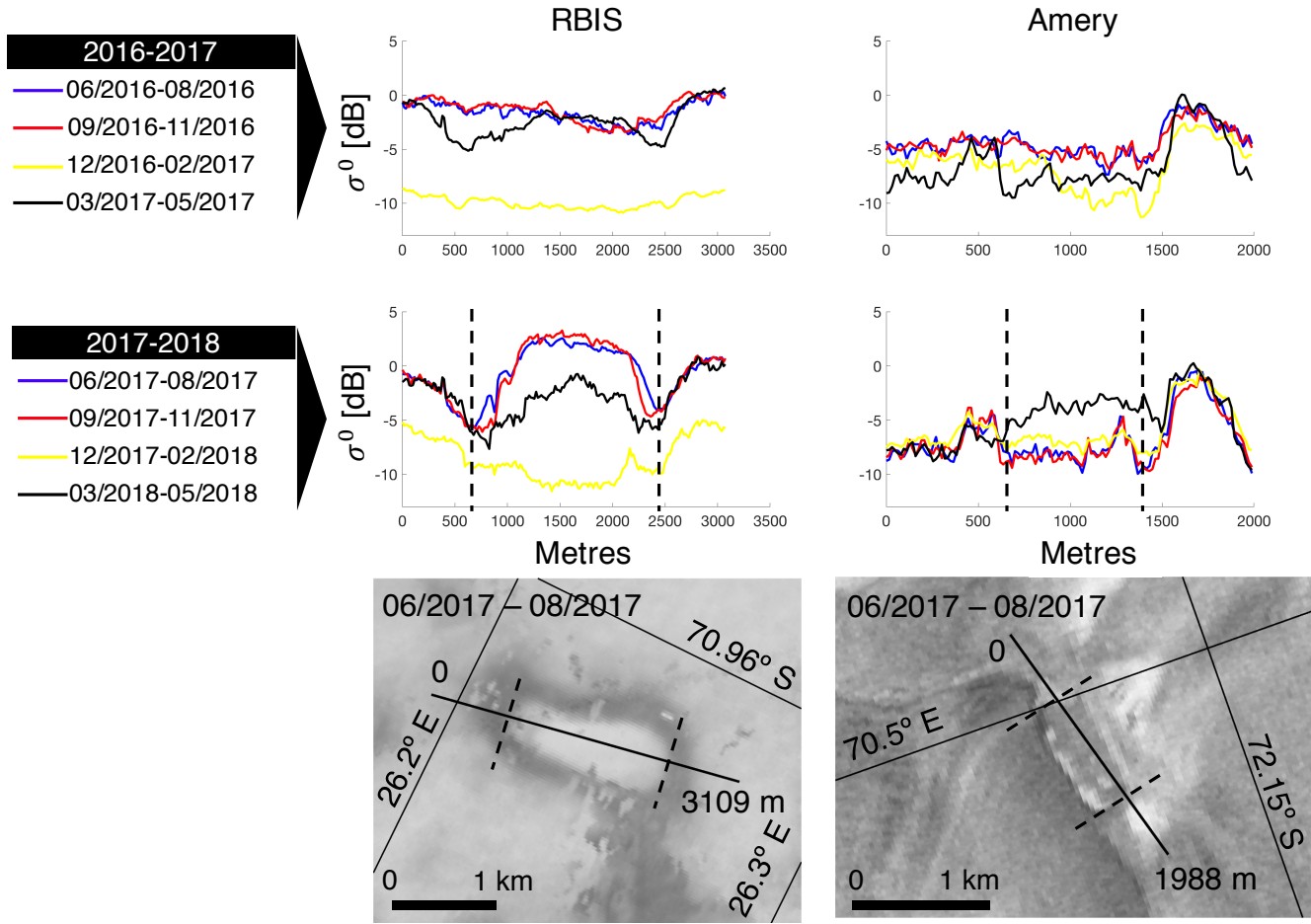

**Figure 4.** Spatial variation of Sentinel-1 $\sigma^0$ for two different lakes: RBIS a (see Fig. 1) on the left and Amery d on the right. The upper and middle panels show the mean backscatter intensities over the lake transect for the June 2016 to May 2017 and for the June 2017 to May 2018 periods respectively. Each curve represents the average $\sigma^0$ over a quarter year of observations. The transects as well as the 2D winter appearance of the feature and its surroundings are illustrated in the bottom panels.

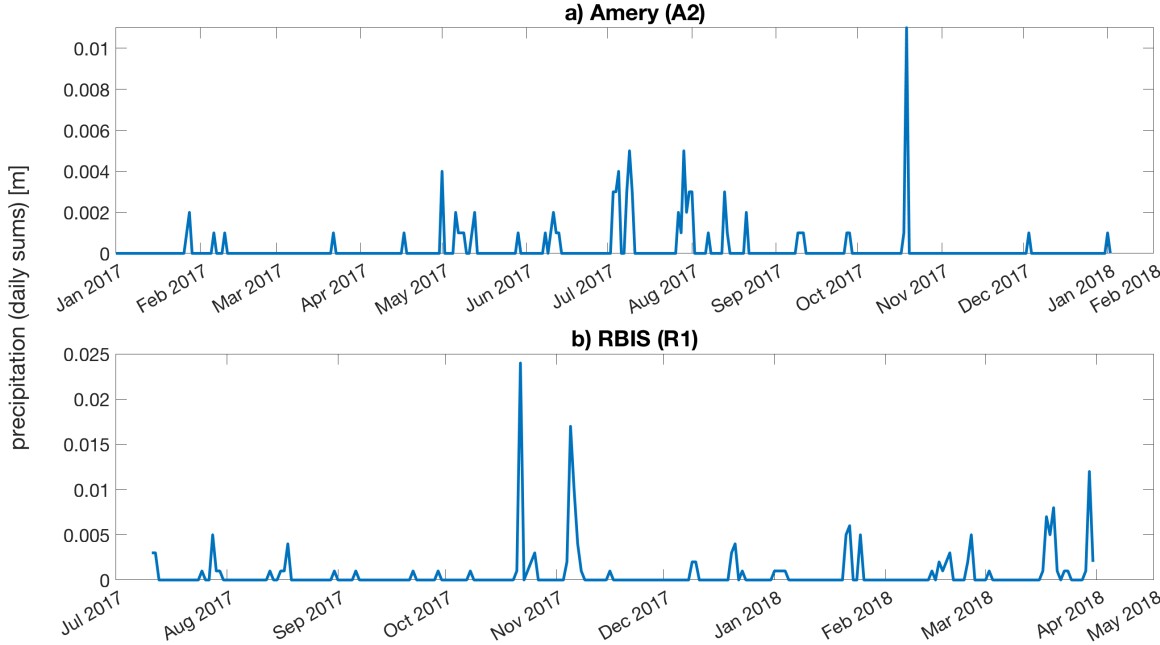

**Figure 5.** Precipitation time series spatially averaged over regions A2 and R1 (shown in Fig. 1) from ERA5, acquired from Google Earth Engine (Gorelick et al., 2017).

6-day revisit cycle allows to get good overall coherence. During summer, low coherence occurs when the surface melts. This can be seen in Fig. 6a and Fig. 6b, where XPGR exceeds the melting threshold in Jan. 2017, and rises towards the melting threshold in Jan. 2018.

On RBIS, the Sentinel-1 data are only available in a 12-day revisit cycle (as in Table 1), which reduces the overall coherence and makes interpretation more complicated as more weather-induced changes in scattering properties could occur. Fig. 5b, for example, shows that region R1 (on RBIS) has a greater amount of total daily precipitation than region A2 (on Amery). Despite the overall lower coherence, the coherence time series on RBIS also show a relatively stable period from August to October, with coherence values above 0.35. Between Oct. 2017 and Jan. 2018, the coherence drops drastically, with an almost null coherence for all surveyed snow, ice and lake areas. The coherence then increases again in February. Overall, snow reaches the highest coherence (0.5–0.6), while the lakes show the lowest coherence.

To better understand the $\sigma^0$ and coherence time series, some representative lake features in the Amery and RBIS zones are analysed in more detail in Fig. 7 and Fig. 8. The outlined lakes on Amery Ice Shelf in Fig. 7 are characterised by dominant blue ice cover with low backscatter intensities, as conveyed by the dark background in the $\sigma^0$ panels. The blue ice region is intermittently covered by a shallow snow layer (e.g. Landsat true colour image of Oct. 2017 in Fig. 7) which decreases in summer (e.g. Landsat true colour image of Jan. 2018 in Fig. 7). This results in a stable ice surface with high coherence values. The lakes, on the other hand, show a more variable behaviour with lower coherence and strong changes in $\sigma^0$ as a

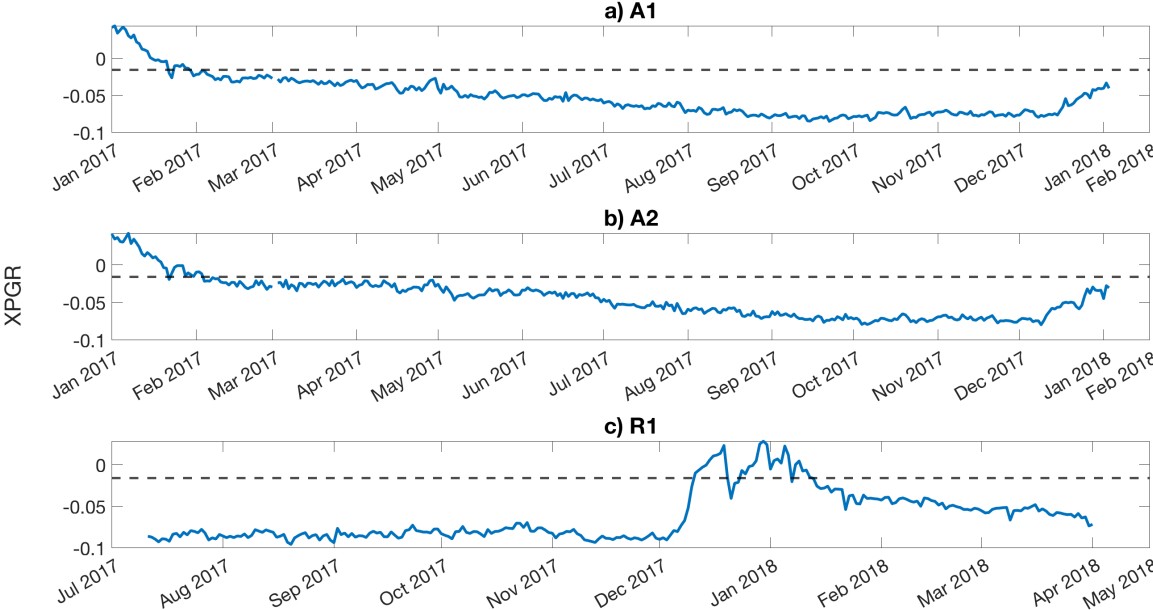

**Figure 6.** XPGR time series calculated with Eq. 1 over regions A1, A2 and R1 (shown in Fig. 1). This time series is an approximation of melt extent time series. The threshold above which the surface is assumed to undergo melt is -0.0158, and is shown as dashed horizontal line in each plot.

result of varying scattering properties. At the end of the summer (Mar. 2017), both lakes show a low $\sigma^0$ whereas it becomes substantially greater in the subsequent acquisitions from July to Nov. 2017. Sigma-nought ($\sigma^0$) and coherence are moreover
not uniform across each lake with the appearance of polygonal features that show large differences between the centre of the lake (with higher $\sigma^0$ and coherence) and a thin strip at the edges (with lower $\sigma^0$ and coherence). This is consistent with earlier observations based on optical satellite imagery, where the lakes show a circular appearance with a thick snow/ice lid in the centre and ice/water at the edges (e.g., Fig. S1 in  Dunmire et al., 2020). This pattern often changes over time, for example, as in the lake Amery c (Fig. 7), where the coherence increases for only half of the lake and not for the other half, which could be an
indicator of gradual, spatially non-uniform refreezing or drainage. One example of such a drainage event could be seen in the small circular feature in the coherence of Amery b in Nov. 2017 (indicated by the arrow in the Nov. 2017 coherence image of Fig. 7), which clearly corresponds to a collapsed circular feature in the Jan. 2018 Landsat imagery. Moreover, between Amery b and c, a hydrological network that is clearly visible as high $\sigma^0$ in the $\sigma^0$ panels is present only in the Mar. 2017 coherence panel as low coherence. This could suggest the surface refreezing between Mar. and Jul. 2017, similar to that discussed by
Antonova et al. (2016).

On RBIS, the lakes are located in an area that contains both snow/firn and blue ice (Lenaerts et al., 2016). In contrast to data acquired over Amery Ice Shelf, the Sentinel-1 SLC acquisition only started in July 2017, with a 12-day revisit (Fig. 8). Lake

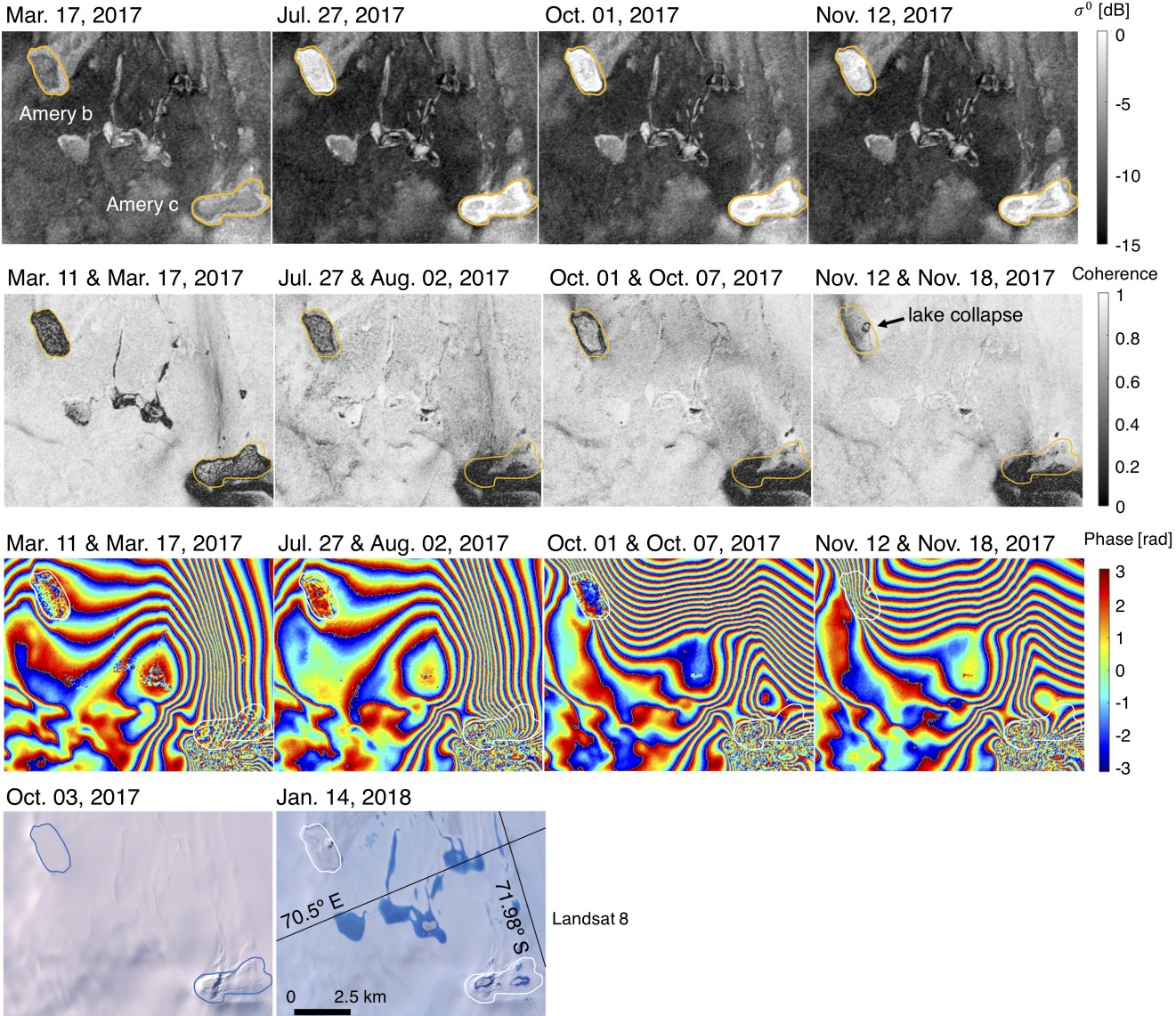

**Figure 7.** Outline for two lakes of interest on Amery Ice Shelf (referred to as Amery b and c in Fig. 1). $\sigma^0$, coherence and resulting phase difference interferograms are shown for four representative dates throughout the year. The high frequency fringes surrounding each lake represent a convolution of both ice flow and tidal motion. Two Landsat true colour images are also shown to aid the visual interpretation of the radar features. The Landsat images are acquired from Google Earth Engine (Gorelick et al., 2017).

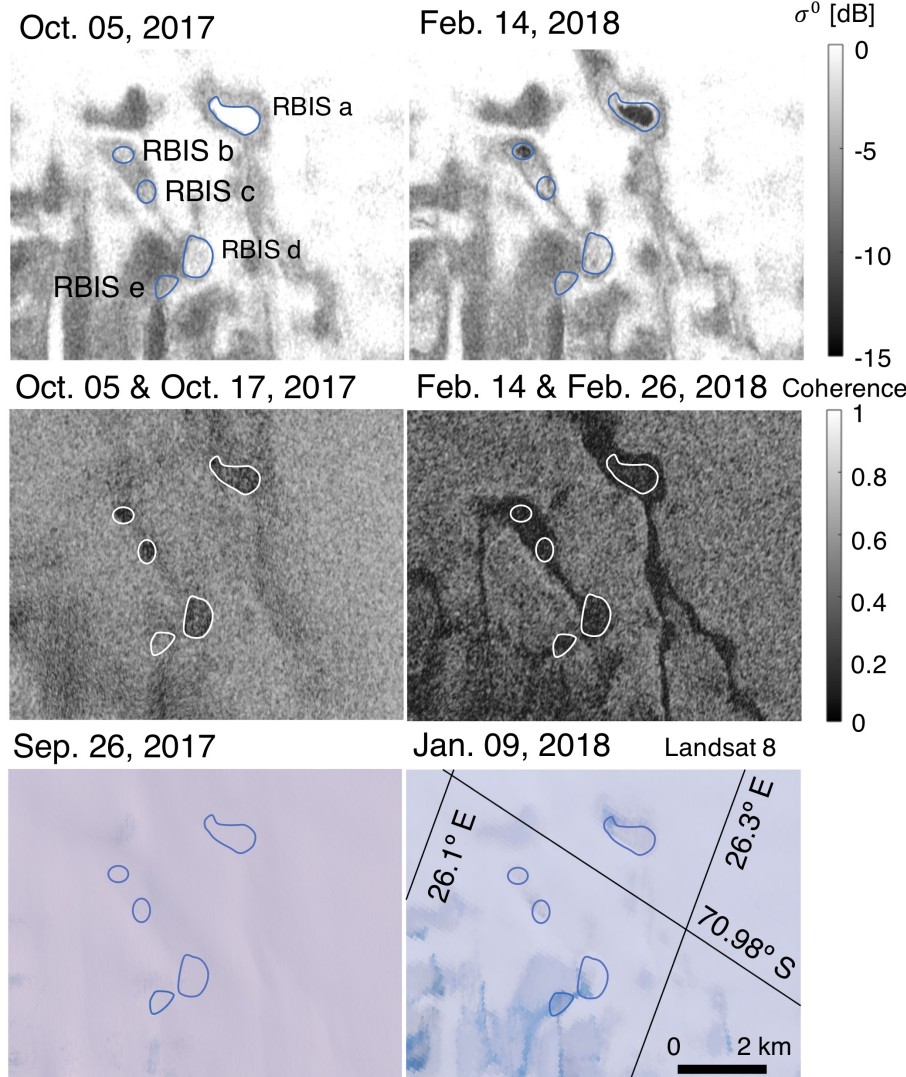

**Figure 8.** Coherence and $\sigma^0$ in the RBIS region before (left panels) and during (right panels) the surface melt in the vicinity of RBIS a in the panels R1 of Fig. 1. These lakes are hereafter referred to as RBIS a to e. Two near-contemporaneous Landsat true colour images are also shown (bottom panels). The Landsat images are acquired from Google Earth Engine (Gorelick et al., 2017).

RBIS a shows a high $\sigma^0$ in October and a low $\sigma^0$ in February, which contrasts with the surroundings. The other lakes show a smaller contrast with their surroundings with only intermediate $\sigma^0$ values. The whole area frequently undergoes coherence losses, especially between Nov. 2017 and Jan. 2018 (Fig. 3d). Fig. 5b shows that precipitation may cause the drop in coherence, as in Oct.–Nov. 2017 it is 2–5 times higher than in other times. Fig. 6c shows that the low coherence between Dec. 2017 and Jan. 2018 may be caused by melt, as the XPGR values during this period exceed the melting threshold. In Oct. 2017 and Feb. 2018, however, coherence is higher (>0.35, see both Fig. 3 and Fig. 8). In both coherence image pairs in Fig. 8, the meltwater features, with low or null coherence values, sharply emerge from the background. In Feb. 2018, the coherence pairs moreover highlight a hydrological connection between the lakes, which is shown as dark curvilinear features between the highlighted lakes in the lower middle panel of Fig. 8. The patterns are clearly newly formed compared to the Oct. 2017 coherence panel of Fig. 8. This feature is not visible in either $\sigma^0$ or optical imagery, highlighting the benefit of InSAR-based coherence for the detection and monitoring of sub-surface lake networks.

### 3.3 Interferogram analysis

Interferometric phase difference maps (Fig. 7) emphasise the differences in spatial cover and melting patterns between the two lakes on Amery Ice Shelf. The centre of lake Amery b is associated with low-frequency fringes in all the acquisitions, even in March, despite the relatively low coherence. Between Mar. 2017 and Oct. 2017, the fringes in the centre of lake Amery b are disconnected from the high-frequency fringes of the surroundings, whereas they connect seamlessly in Nov. 2017. This pattern of discontinuity is consistent with lower coherence at the edges of lake Amery b, which follows the orange delineation curve in the Oct. 2017 coherence panel of Fig. 7. That both fringe discontinuity and coherence increase through time indicates the presence of the lake until Oct. 2017, followed by a lake refreeze or drainage in November of that year. Consistent with our InSAR-based observations, Landsat images show a smooth snow-covered surface in Oct. 2017 and a rough doline-like surface in Jan. 2018 (labelled as lake collapse in the coherence panel of Fig. 7). This supports the hypothesis that the lake drained and the surface collapsed, and highlights the potential of coherence and interferogram for analysing meltwater dynamics.

On the eastern part of RBIS, the interferogram shows a different potential for analysing meltwater dynamics (Fig. 9) as it shows a phase reversal from right to left of the Dec. 2017 phase image (i.e. fringes change from red–blue–green–yellow to red–yellow–green–blue, forming a concentric pattern of deformation associated with a series of dense, closely spaced fringes) compared to a continuous phase from right to left of the Apr. 2018 image (i.e. fringes are constantly red–yellow–green–blue). This phase reversal indicates that the lake has a displacement in the satellite line-of-sight which is opposite to the rest of the ice shelf. As the ice shelf background fringes correspond to the ice flow and presumably tidal component, in this case moving away from the satellite line-of-sight, the lake fringes indicate an uplift as a result of ice shelf rebounce after lake collapse. This hypothesis is consistent with earlier observations, including the rebound effects described by Banwell et al. (2013). Indirect indicators of this lake collapse can also be observed in the Landsat 8 images before/after the collapse, as the roughness of the surface strongly increased after the collapse. By counting the fringes, the feature consists of approximately 7 fringes, each measuring 2.8 cm in the line-of-sight. Assuming a vertical movement, this corresponds to an uplift of approximately 24 cm

(taking into account an incidence angle of approximately 35°). However, this amount of uplift is only an approximation of a displacement relative to ice flow and tidal component, and needs in situ observations to validate.

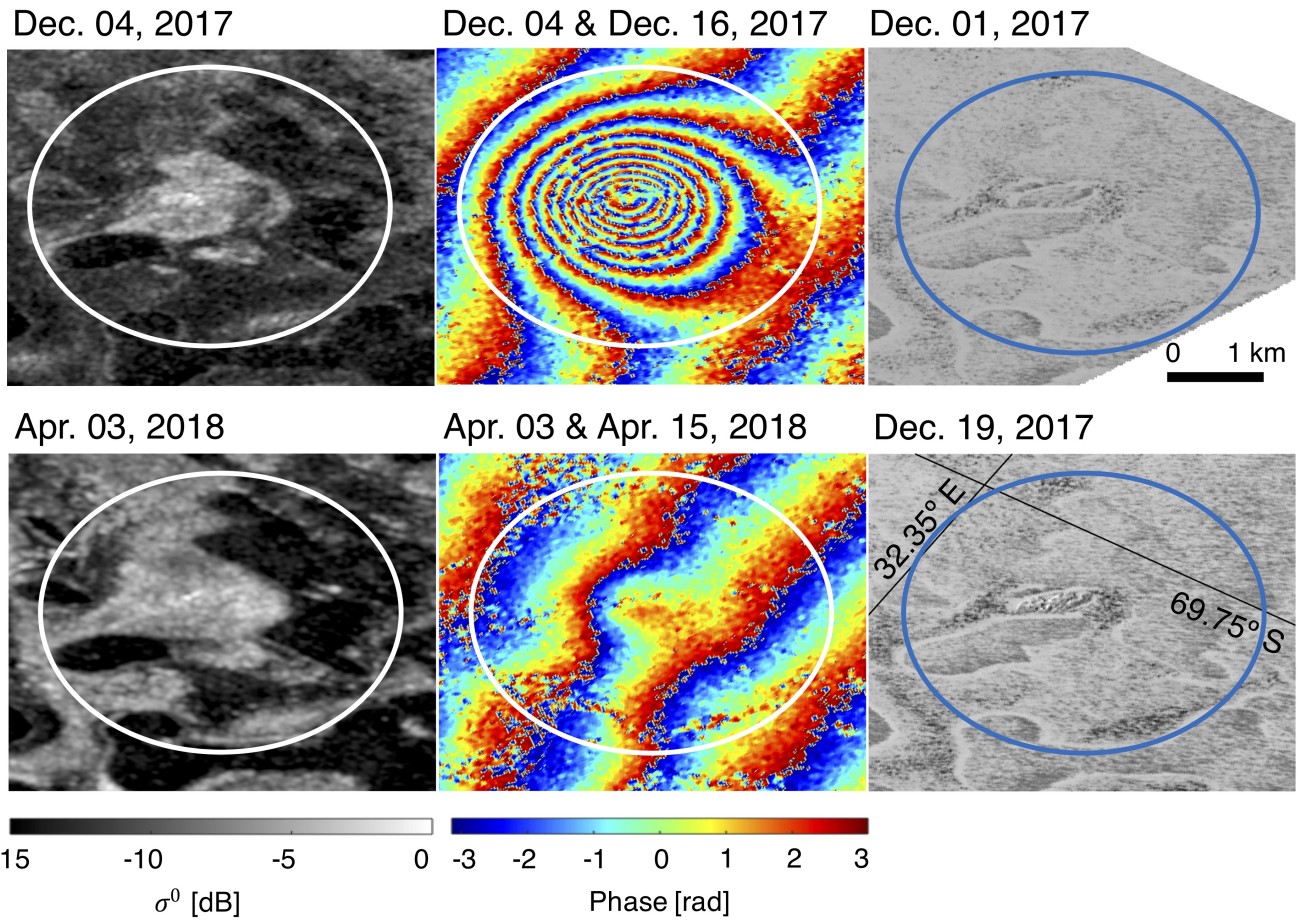

**Figure 9.** Sigma-nought ($\sigma^0$) and interferograms for a lake feature in the east of the RBIS experiencing drainage in December 2017 (upper panels) and ice-cover collapse in April 2018 (lower panels). Two near-contemporaneous Landsat 8 panchromatic (band 8) images are also shown (right panels). The Landsat images are acquired from Google Earth Engine (Gorelick et al., 2017).

Another potential of interferogram time series is the detection of lake refreezing, as can be observed for the large lake feature in the middle of Amery Ice Shelf, labelled as Amery a in Fig. 1. Both Amery a and the surrounding ice shelf show an overall low $\sigma^0$ on Jan. 4, 2017 (Fig. 10). In subsequent weeks, the $\sigma^0$ of the snow surrounding area increases. The $\sigma^0$ of the lake gradually increases between Jan. 4 and Mar. 17, 2017 and gradually decreases between Mar. 17 and Apr. 22, 2017. The low $\sigma^0$ on Jan. 4, 2017 is likely a result of surface melt, while the subsequent rise in $\sigma^0$ is likely due to refreezing. This pattern corresponds closely with the refreezing pattern identified by Spergel et al. (2021), who also identified a gradual refreezing towards the centre of the lake over 66 days based on transition from high-to-low backscatter intensity only. However, compared to interpreting the

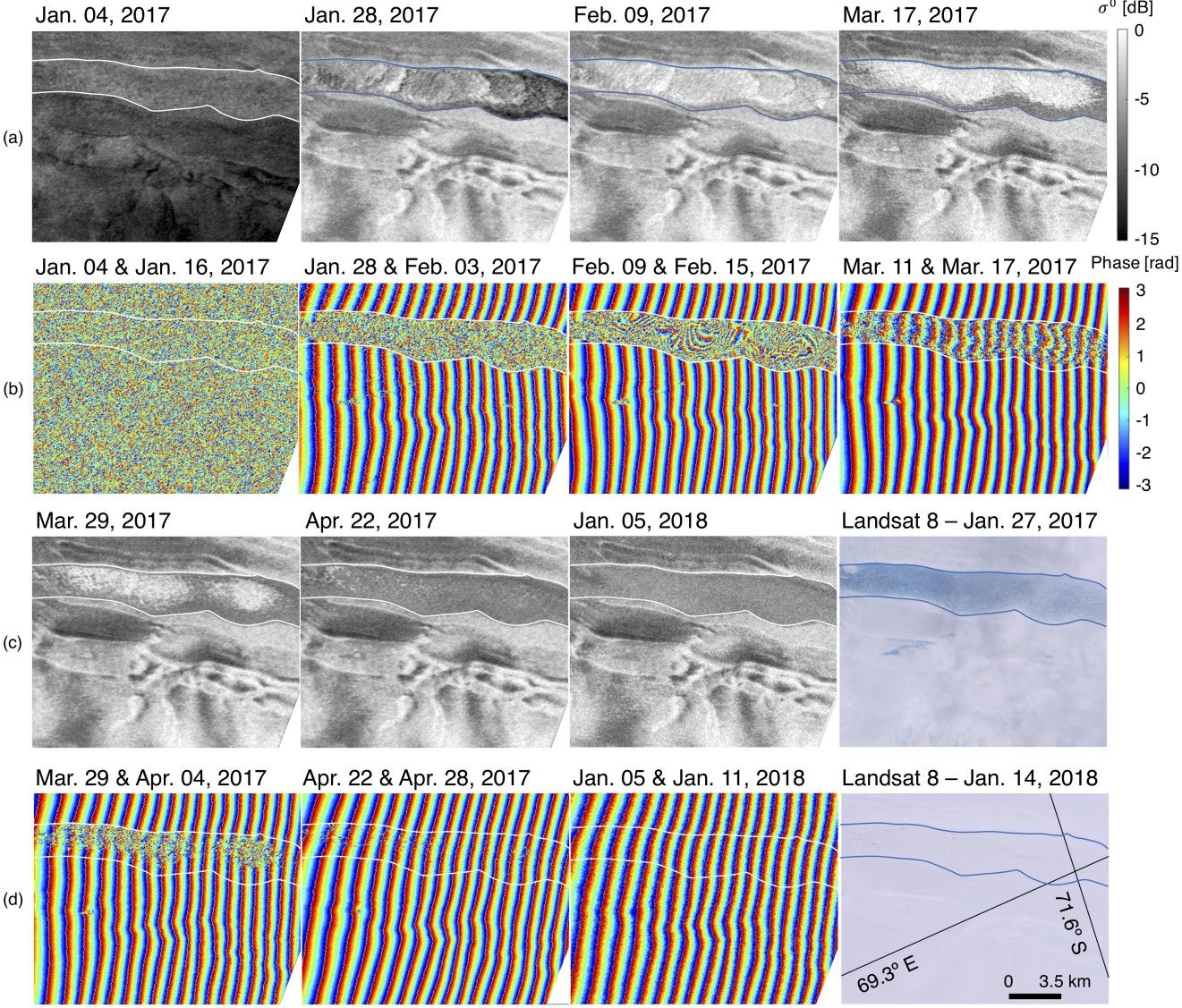

**Figure 10.** Sigma-nought ($\sigma^0$) and interferograms for a large lake in the middle of the Amery region experiencing a refreezing process. $\sigma^0$ refers to the first image of the interferogram pairs. The interferograms clearly show that the lake refreezing occurs in the first half of 2017. The lake remains frozen throughout the remainder of 2017 and in 2018. The two Landsat true colour images in the lower right corner provide a visual evaluation. The Landsat images are acquired from Google Earth Engine (Gorelick et al., 2017).

refreezing of the lake solely based on backscatter intensity, adding interferograms to the observation helps reduce ambiguities in the interpretation. The interferogram of both Amery a and the surroundings is completely incoherent on Jan. 4, 2017, which can be interpreted as changing scattering properties due to intense surface melt. Since Jan. 28, 2017, the coherence of the snow surrounding area increases due to refreezing, as can be seen from the visible regular fringes. For the lake, however, this increase in coherence lags behind and only recovers slowly as more portions of the lake start to refreeze. During the refreezing, the fringes patterns over the lake gradually recover while the incoherent noise gradually diminishes. Both the Landsat panels of Fig. 10 and Fig. 5a show that Jan. 2017 is a more intense melt season than Jan. 2018, which is consistent to the observation from the fringes.

## 4 Discussion

Using SAR-based observations acquired across two East Antarctic ice shelves, this study presents evidence of the utility of backscatter intensity and coherence to assess meltwater lake dynamics. Low backscatter intensities can indicate blue ice areas or strong absorption due to meltwater, while high backscatter intensities indicate rough surfaces or strong volume scattering due to larger refrozen snow grains. Moreover, the partly frozen lakes often show a bright centre (high $\sigma^0$) that can be attributed to the single bounce mechanism at the rough ice–water boundary (Engram et al., 2013; Atwood et al., 2015; Antonova et al., 2016). Due to this contrasting behaviour, the identification and characterisation of the meltwater features based only on backscatter intensity are not straightforward. Several of the observed lakes, for example, show $\sigma^0$ similar to their surroundings for long periods, and even during the freezing/melting processes (e.g. Fig. 8 and Fig. 10).

Backscatter intensity may not therefore be sufficient to fully characterise meltwater processes. Interferometric coherence, however, provides additional dynamic information as it helps assess the degree of stability of the ice cover between two acquisitions. Coherence is an important property estimated from interferometric computation of SLC data. For repeat-pass acquisition, a loss of coherence mainly reveals the extent of a surface change (Zebker and Villasenor, 1992). However, with substantial microwave penetration depths in snow/firn, coherence variations can indicate changes in scattering properties. Coherence losses may consequently be due to changes in volume scattering (Zebker and Hoen, 2000) or subsurface processes. Low coherence between interferometric images can therefore indicate altering scattering properties (e.g. a strong snowfall or an intense melt event), but also changes in ice–water interface due to refreezing meltwater lakes (Antonova et al., 2016) where refreezing may result in a gradual increase in coherence. Ice and snow areas are typically characterised by a high coherence, while meltwater lakes show a low coherence due to the constantly changing ice–water interface and the increased attenuation due to the presence of water. This added value of coherence is shown, for example, in Fig. 7 and 8, where coherence provides more insight into the temporal dynamics of the lakes than the $\sigma^0$ images alone. The change from complete polygonal low coherence patterns to partly high coherence (Fig. 7), for example, provides an important indicator of the gradual refreezing patterns (i.e more refreezing in the centre than at the edges). These results correspond to the study of Antonova et al. (2016), where the melting and refreezing of lake ice could be observed by using both backscatter intensity and coherence image time series.

Beyond coherence, we also demonstrate the potential of interferometric phase for assessing meltwater dynamics in areas of high coherence. For example, the deformation due to rapid meltwater events, such as drainage and collapse, may be captured, if the fringe pattern in the lake area appears highly distinct to the surroundings affected by tidal and horizontal motion. Within this context, we identify two advantages of phase fringes over $\sigma^0$ and coherence alone: i) an easier detection of stable ice and lake refreezing than coherence and backscatter intensity and ii) the detection of relative motion due to uplift and subsidence events as a result of lake drainage or lake filling. The first advantage is clear in Figs. 7–10, where the phase patterns allow additional interpretation of the refreezing patterns which cannot be imaged by coherence or backscatter intensity alone. The second advantage is shown clearly in Fig. 9, where the closely spaced fringes shown could be used to estimate the presence of an uplift event due to drainage.

While InSAR-based techniques show clear potential for monitoring meltwater lake evolution, there are several key limitations associated with this technique compared with conventional optical- and SAR backscatter-based imaging. First, it requires high coherence between image pairs to allow a meaningful interpretation of meltwater lake dynamics (e.g. as in Fig. 10). When the revisit cycle for SLC data is long or when the surface changes due to other processes (e.g. strong snowfall events, as shown in Fig. 5) are frequent, the interpretation of coherence and phase changes can be limited. On Amery Ice Shelf, the Sentinel-1 mission has a 6-day revisit, whilst the revisit period on RBIS is 12 days. The amount of precipitation is also lower on Amery Ice Shelf compared to RBIS. Due to these differing imaging times and weather, the lake processes are better observed on Amery Ice Shelf than RBIS. Second, the interpretation of phase change should be performed relative to the displacement of the lake surroundings in the line-of-sight. For example, as the meltwater lakes typically develop in locations with strong ice and/or tidal displacement, interpretation should be done relative to that displacement. Therefore, to better derive the exact height change of lake ice lids, additional processing will likely be needed to cancel out, for example, the effects of ice-shelf flow (Mohajerani et al., 2021) and to filter out signals due to tidal movements (McMillan et al., 2012). With SAR acquisitions from sensors in both ascending and descending orbits, it is however possible to better quantify the lake subsidence/uplift.

A potential improvement of lake monitoring using InSAR is the launch of new satellite missions. The launch of Sentinel-1C (Torres et al., 2017), for example, can provide <6-day imaging capabilities to improve coherence of the ice and snow surface. The launch of the NASA-ISRO SAR (NISAR) mission, moreover, provides L-band and S-band repeat-pass interferometry with the repeat cycle of 12 days (Rosen et al., 2017). The long wavelength of this mission has the potential to measure deeper lake dynamics and to circumvent drifting snow and other atmospheric effects.

## 5 Conclusions

This study has provided insights into the utility of InSAR for monitoring meltwater lake dynamics on ice shelves. Four Antarctic ice shelf regions subject to intense summertime melt have been analysed using Sentinel-1A/B C-band SAR imagery, corresponding available Landsat 8 imagery, ERA5 precipitation data and SSMIS brightness temperature data. The spatial and temporal inspection of the meltwater features conveys that backscatter intensity allows identification of freezing and melting events, as the lakes show an increase of the backscatter intensity due to the water–ice boundary when the lake is not com-

pletely frozen. The extent of such dynamics depends on the morphology of the lake and on the weather conditions. We show that meltwater detection using backscatter is, however, not straightforward, as meltwater lakes often show similar backscatter intensity values to their surroundings. In such circumstances, InSAR information can be useful to increase the confidence of such delineation, especially during the freezing and melting period. In addition, we show that InSAR-derived information can also be used to observe meltwater lake evolution (and potential drainage) with high accuracy beyond that afforded by conventional backscatter or optical satellite imaging. Specifically, InSAR coherence information allows for the detection of changes in the ice–water interface, which shows clearer patterns than the backscatter intensity alone, while interferometric phase can effectively track the spatial and temporal evolution of ice refreezing. Maps of interferometric phase moreover allow for the detection of abrupt lake drainage (or filling) events via changes in the relative displacement of the surface between successive SAR passes.

Despite noted limitations to current Sentinel-1 InSAR imaging over parts of Antarctica, this study shows that InSAR provides promising potential for monitoring meltwater lake dynamics beyond that afforded by conventional, backscatter-only, analyses. Such potential could pave the way for dedicated Sentinel-1 meltwater products that could facilitate the study of ice shelves in a changing climate.

*Code availability.* The DORIS software used to process Sentinel-1 SLC data is available at http://doris.tudelft.nl.

*Data availability.* The TanDEM-X data used for geo-coding the InSAR SLC products on the RBIS are available at https://doi.org/10.1594/pangaea.868109.

*Author contributions.* SL developed the idea of this study and provided access to the mapped locations of the meltwater ponds on the RBIS and TanDEM-X data. PLD provided expertise in processing and interpreting InSAR data. WL was responsible for managing the data, processing the data with DORIS, generating melt extent and precipitation time series, processing and analysing the results, producing the figures, and providing the manuscript.

*Competing interests.* The authors declare they have no conflict of interest.

*Acknowledgements.* We are grateful for the DORIS development team at TU Delft. We acknowledge Copernicus Open Access Hub for providing Sentinel-1 SLC data, National Snow and Ice Data Center (NSIDC) for providing the SSMIS brightness temperature data, and Google Earth Engine for providing Landsat 8 and Sentinel-1 GRD images. This study contains modified Copernicus Climate Change Service information.

DEM for visualisation is provided by the Byrd Polar and Climate Research Center and the Polar Geospatial Center under NSF-OPP awards 1543501, 1810976, 1542736, 1559691, 1043681, 1541332, 0753663, 1548562, 1238993 and NASA award NNX10AN61G. Computer time is provided through a Blue Waters Innovation Initiative. The DEM is produced using data from DigitalGlobe, Inc.

We would also like to thank Dr Lorenzo Iannini and Malte Manne for proof-reading and discussion. Additionally, we thank the Referees for reviewing, and Dr Chris Derksen for editing the manuscript.

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
