# Peer review of "The potential of synthetic aperture radar interferometry for assessing meltwater lake dynamics on Antarctic ice shelves"

_The Cryosphere, 2021_

## Referee Comment (RC1)

**Review of "*The potential of InSAR for assessing meltwater lake dynamics on Antarctic ice shelves*" by Weiran Li et al.**

Li et al. assess the utility of synthetic aperture radar interferometric (InSAR) techniques (coherence, phase) to track the seasonal evolution of supraglacial lakes across two East Antarctic ice shelves. While similar techniques have previously been applied to Arctic regions (e.g. Antonova et al., 2016), this study presents the first application of C-band InSAR observations for lake monitoring in the Antarctic. Although the use of InSAR is found to be limited in summer due to extensive surface melting (resulting in phase decorrelation), the authors show that, compared to backscatter (and supplementary optical imagery) alone, coherence and phase information can provide important additional insights into the exact dimensions and timing of lake evolution during non-summer months. These insights have importance for understanding the processes behind ice-shelf weakening (and potential destabilization) in a changing climate, with implications for global sea-level rise. In this regard, I believe that the results presented in this manuscript will be of interest to the general readership of The Cryosphere although, prior to publication, I recommend moderate-to-major revisions. I outline my reasoning for this decision in the general comments below.

**General Comments**

My main concern pertains to the overall readability of the manuscript. While it is clear that the authors understand the background literature, methods, key results and implications of their research, their writing style is in general unconventional for a piece of scientific writing, insofar as it is highly verbose, often grammatically and/or typographically incorrect and, hence, difficult to follow/comprehend from a reader's point of view. This is further confounded by what appears to be inconsistencies in the clarity/style of writing adopted in different sections of the manuscript, missing information needed to fully understand the datasets and/or logic of arguments presented, and the occasional lack of relevant citations throughout the text (see my specific and technical comments for further details).

To address these issues, I recommend that all coauthors take the time to carefully restructure the wording of the manuscript to: a) more logically explain (and justify fully) the choice of all techniques and methodological decisions used/taken, b) correct typographical/grammatical errors and, c), cut down and hence improve the overall focus/narrative of the text. To assist the authors in this regard, I have made some suggestions on how the first two paragraphs of the introduction could be rewritten (see bottom of this document). If needed, the authors may also find the following resource (https://www.the-cryosphere.net/submission.html#english) and links therein helpful.

**Specific Comments**

**Title.** 'InSAR' is an abbreviation and hence inappropriate for use in a title. Suggest rephrasing title to: 'The potential of synthetic aperture radar interferometry for assessing meltwater lake dynamics on Antarctic ice shelves' or similar instead. (see also my comments regarding Line 246).

**Line 2.** Suggest replacing 'Yet' with 'Despite these phenomena'. Replace 'or' with 'and'.

**L3.** Suggest either ending sentence after 'limited', or briefly explaining what the limitations of optical satellite imagery are here.

**L8.** Change 'The analysis' to 'Our analysis'. At end of this sentence, change 'confounded' to either 'hard to distinguish' or 'indistinguishable'. Then change next sentence to: 'Despite this

finding, we show using a combination of backscatter and InSAR observations that lake dynamics can be effectively captured during other non-summertime months'.

**L11.** Sentence beginning 'In particular'. For conciseness, suggest merging this and next sentence to: 'Moreover, our findings highlight the utility of InSAR-based observations for discriminating between refrozen ice and subsurface meltwater, and indicate the potential for phase-based detection and monitoring of rapid meltwater drainage events'.

**Introduction**. While this introduction is well researched and all of the key information/literature is there, my sense is that it is rather verbose and/or repetitive and could be shortened considerably. For readers with little/no remote sensing expertise, I'm also a little concerned that there are multiple logic gaps which would make this difficult to follow. See my suggested rewrite of the first two paragraphs (bottom of document) to get a sense of how this could be rewritten for brevity and clarity.

**L56.** Remove 'The' and begin sentence 'Coherence is considered an indicator of changes'. Note also that phase difference does not correspond to the 'average' difference, but a very precise measurement of whole wavelength (and some fractional component) range difference. Suggest editing the rest of the sentence to reflect this.

**L59.** Sentence beginning 'This combination'. Who expects this? Add a reference to back up this claim, otherwise say 'We expect this' or similar.

**L67.** Suggest changing to: "However, the value added using InSAR for such applications has not yet been examined".

**L68-71.** The structure of this paragraph is rather difficult to follow. Suggest beginning with "In this paper, we assess the potential of C-band backscatter and InSAR data to … For this purpose, we use a combination of backscatter, coherence and phase information to monitor recent meltwater features over two East Antarctic locartions – the Amery and Roi Baudouin (RBIS) ice shelves – using data collected by Sentinel-1a/b in 2017/2018. To supplement the interpretation of our (In)SAR-based analyses, we also utilize spatially and temporally collocated optical satellite data.

**L75**. Sentence is very long and could be split in two after reference to Lenaerts et al. (2016).

**L81.** Why was a lake dataset not available? The reader shouldn't need to guess this, so state explicitly here. (I think you mean that no previously published dataset exists?). When mentioning Landsat data, also point readers to Section 2.2 for reference.

**L83.** Sentence beginning 'Our lake class'. If my understanding of the above is correct, then this sentence is confusing as it suggests that a preexisting lake dataset does indeed exist. If so, why didn't you use that here? Please either edit sentence of clarify or remove from the text.

**Figure 1.** This is a nice figure, but please add latitude/longitude information to each panel so that readers can easily deduce locations. Please also consider showing two additional, zoomed out panels showing the location of each subset within both ice shelves, and (if necessary) in these and all pre-existing panels, add the ice sheet grounding line for reference (e.g. https://doi.org/10.7280/D1VD6G). (I know that the lower right panel shows this to a certain extent, but it's difficult to see details. In general, I also find the labels rather small and difficult to locate, so these could also be enlarged (plenty of space on figures to do this).

**Figure 1 caption**. Please define RBIS in full in caption. Replace 'close ups' with 'inset' or 'detail'. Insert comma after 'panels'. Change 'delineated in black curves' to 'delineated as black

curves'. In next sentence, what does 'indices' refer to? Labels? If so use 'labels' instead for clarity. Suggest rephrasing following sentence to read: "…are also delineated for comparison against backscatter intensity and coherence values observed over lakes (Fig. 2)' or similar. For general readability, next sentences could/should read: 'Panel R2 illustrates the lake feature shown in Figure [insert number here]. Inset shows location of the analyzed locations'. Please also state in caption which band/band combinations are shown (Landsat).

**Table 1 caption**. Should read '… used in this study'.

**L99**. Please add more information (and references if necessary) on how the images were denoised, calibrated and corrected here. (The reader shouldn't have to look it up for themselves).

**L92**. I presume this is a typo and should say 10x10 m resolution (i.e. the native resolution of IW GRDH; https://sentinels.copernicus.eu/web/sentinel/technical-guides/sentinel-1-sar/products-algorithms/level-1-algorithms/ground-range-detected/iw)? If not, then how does this impact the rest of your backscatter analyses?

Also, if my understanding is correct then NRCS is identical to radiometrically calibrated, sigma-nought ($\sigma^0$) backscatter imagery. Sigma-nought backscatter is the much more commonly adopted term in the geosciences (RCS more so in engineering), so given the likely readership of The Cryosphere, I would instead refer to $\sigma^0$ in place of NCRS here and universally throughout the manuscript. Following this point, be careful that the imagery you downloaded from Google Earth Engine isn't already in sigma-nought format, as the 'radiometric calibration' you mention GEE perform above may imply.

**L95.** Confusing. If you have multi-looked the SLC to create your own GRD/Sigma-nought imagery then why did you bother downloading GRD imagery from GEE? Clarify here.

**L97.** Why is a geoid only used for Amery? I expect this is a typo?

**L98+** This is a good example of missing methods I mentioned in my general comments. Here, I am surprised to see absolutely no information about how the authors generated their interferograms. This must be added, including, as a minimum, information on e.g. temporal/perpendicular baselines used, type of processing performed (I assume single-pass DInSAR?) and (if used) any DEM used to remove topographical phase.

**L99-100.** Again, for clarity/reproducibility purposes, much more info should be included here on e.g. the band/band combinations used, pixel resolutions of the data, and any relevant pre-processing steps you applied to your Landsat imagery. Stating that you simply downloaded them off GEE is not appropriate for a scientific paper.

**L105.** I think it's important to explicitly state here that for each class type analyzed, you calculated the mean and standard deviation backscatter for all observed features. As written this is not obvious, and leads to confusion over what the difference is between 'lake class' and the 'individual lakes' (for this reader at least …). This could also be made more explicit in the caption of Figure 2.

**L106.** For clarity, 'mono-dimensional' should be changed to 'cross-sectional'.

**L107.** Repetition of coherence, NCRS and phase. Suggest restructuring sentence to avoid this.

**L112.** Suggest changing to: 'The mean sigma-nought timeseries of lakes, snow and ice (cf. Section 2.2) display strong seasonal variability, consistent with the changing nature of both

surface snow and ice properties and the evolution of supraglacial lakes through time (Figure 2). On Amery Ice Shelf, our observations reveal…'.

**L13.** This is written in an odd manner which implies that snow transforms into lakes and then ice, which is not what the authors intend to say. What I think they mean to say is that snow, lakes and ice for the most part display different (though reasonably constant) backscatter properties throughout the year, with the exception of JF when the backscatter associated with snow and lakes fall rapidly.

**L116-125.** To shorten the text here, I question whether the authors even need to discuss (and, in Figure 2, show) the individual lakes because for the most part, the average of multiple mapped lakes makes seems to support their arguments just as well. In this regard the individual lake observations are a slight distraction from the overall story revealed by the class averages, so I think they could probably be removed. (as the authors show, there is significant variability from one lake to the next, so focusing in on specific lakes only serves to deviate from what's happening on the whole). This is also largely true for the coherence discussion of RBIS a and f in Section 3.2, as your later coherence images (Figure 7) in any case demonstrate the process of refreezing in a much more convincing way).

**L126-134.** For clarity, suggest editing sentence to say '…of select cross-sectional transects. In the case of both RBIS 'a' and Amery 'd' (location shown in Figure 1), for example, backscatter timeseries show significant inter-annual variation (Figure 3)'.

**L129**. 'Border of low NRCS and inner areas of high NRCS'. Revise this sentence to explicitly state that this refers to the edge and central regions of the lake, respectively.

**L136-152.** These are clear, well written paragraphs. They are logical and concise, and could be considered a model for how the rest of the manuscript should be written.

**Figure 2.** See my comments regarding L116-125 above. If the authors choose to retain the analysis of individual lake features, then they should make the lines thicker as these are currently very difficult to see both on-screen and in print. To enhance visibility of these lines, I'd also consider making the standard deviation ribbons more transparent as these currently dominate/clutter the figure. As per Figure 1, I also think the labels (and legend especially) should be made bigger/more prominent.

For the coherence plots (and to a lesser degree sigma-nought), I wonder if the high frequency variability discussed by the authors could be smoothed out using something like a running mean? While this variability is interesting, it's a little distracting, and is later largely ignored in the text anyway.

**Figure 2 caption.** Sentence should read '… over the Amery and Roi Baudouin ice shelves (see Fig. 1 for locations). Change 'Moments …' with 'Times with a lack of 6/12-day…'.

**Figure 3.** As per Figs 1 and 3, please make all labels larger. Please also add lat/longs to both maps along with scalebars. Also consider zooming both images to show more detail over lakes (bottom right panel especially).

**Figure 3 caption.** Move '(see Fig. 1)' to end of sentence, and change to '(see Fig. 1 for locations). Add comma before 'respectively'. Next sentence should also say '… over a three-month period'. In next sentence, should read: '…of the feature and its surroundings …'.

**Figure 4.** Nice figure! As above regarding label size.

**Figure 4 caption.** Remove 'synoptic' (incorrect usage in this context) in first sentence, and cross-reference Fig. 1 for location at end. In next sentence, remove 'the' proceeding

coherence, and add '… and resulting phase difference interferograms are shown …'. In next sentence, should say 'The high frequency fringes surrounding each lake represent a convolution of both ice flow and tidal motion'. In the last sentence, please state which band/band combinations are shown (Landsat).

**Figure 5.** Really nice figure, but please add lat/longs and scalebars.

**Figure 5 caption.** Unnecessary use of 'right' which should be removed. In the next sentence, change 'hereby' to 'hereafter'. In the following sentence, change 'reported' to 'shown (right panels)'. Please also state which band/band combinations are shown.

**L147.** I think this should say 'between Oct. 2017 and Jan. 2018'. Change 'polygons' to 'surveyed snow, ice and lake areas'.

**L154.** Amery Ice Shelf.

**L159.** Replace 'brighter' with 'greater'.

**L172.** Why 'possibly'? Provide evidence to justify claims here. (Also, RBIS is a rather slow flowing ice shelf, so horizontal displacement should not influence phase coherence over 12 days as much as one might think (see Mohajerani et al. (2021) who were able to map GLs across this region Antarctica with good coverage using double difference InSAR. This technique requires almost perfect coherence, suggesting 12 days is more than sufficient here).

In the next sentence (beginning 'In Oct. 17…'), I think better referencing to Figures 2 and 5 is needed as I don't see any change in coherence from Figure 5 alone.

**L176.** Change tense to be consistent with the rest of the paragraph. Also, while what you go on to say in Lines 176-177 is technically true, visually I can't tell the difference between the lakes you are discussing and the drainage network. Suggest rewriting this sentence for clarity to specifically emphasize the observed change from a lake to a (presumably) connected drainage network through time.

This is a really nice observation by the way, demonstrating in a compelling manner the utility of coherence to see what simple optical and/or backscatter images cannot.

**L179.** Suggest rewriting to begin: 'Interferometric phase difference maps (Figure 4) emphasize… Amery ice Shelf.

**L180.** Initially I didn't see any fringes you refer to (c/w Amery c for example) given the dominance of the high frequency (ice flow) fringes surrounding the lakes, but then I realized you meant the very low frequency fringes on the lakes themselves (~1 cycle of -π to π only). I suggest you state this more clearly (and perhaps label the figure accordingly) so that readers don't incorrectly focus in on the high frequency fringes.

**L182.** Edges of what? I can work out what you mean, but this can be written more clearly for ease of reading. Possibly also consider citing appropriate figures and panels.

**L183.** Suggest writing as '…... increase through time indicates the presence of lakes until October 2017, followed… in November of that year. Consistent with our InSAR-based observations Landsat …'.

**L179-203.** In general, this is another clearly written and easy to comprehend series of paragraphs compared with the earlier section of the manuscript.

**L179-186.** Regarding Amery Ice Shelf, what (if anything) can we learn about the detection of the hydrological network that is clearly visible in Figure 4 (top row), and which disappears after

March 2017 in the coherence images? (suggesting formation between Mar 11 and 17th and persistent presence (freezing?) thereafter). This is a visually striking feature in the center of these panels that I was surprised to see no discussion of here and/or in Section 3.2.

**L187.** I'm not quite sure I follow this, as the color scheme always goes from blue to blue. Suggested rewriting for clarity.

**L198.** How big was this uplift? I think that would be a valuable addition here, and can be estimated either through unwrapping the phase or counting the fringes.

**L190.** And presumably some tidal component, as *I think* tides haven't been removed? (see also my comments on the omission of any methods detailing exact InSAR processing above).

**L194-203.** Great series of observations.

**Figure 6.** Nice figure, but please add lat/longs and scalebars to all panels.

**Figure 6 caption**. Replace 'interferometric phases' with 'interferometric phase'. For brevity, suggest rewording next sentence as 'Two near-contemporaneous Landsat 8 panchromatic (band 8) images are also shown (right panels)'.

**L201.** Replace 'starting at the edges' with 'towards the center of the lake' or similar.

**Figure 7**. Very nice series of observations! Enlarge labels and add lat/longs and scalebar, though.

**Figure 7 caption.** Replace 'interferogram phases' with 'interferograms' or similar. Please also state which band/band combinations are shown for Landsat imagery.

**L204.** I'm not sure I completely follow what you're trying to say here, as the sentence contains a grammatical error. Suggest rephrasing for greater clarity.

**L205.** Suggest beginning this section like: 'Using SAR-based observations acquired across two East Antarctic ice shelves, we present evidence of the utility of backscatter …'.

**L213.** Change 'Coherence' to 'Interferometric coherence'.

**L215.** And all other types of SAR SLC data … not just that acquired by Sentinel-1.

**L222**. I think its important to stress here that low coherence isn't just about refreezing (or not). Radar waves are fully attenuated by water, so you will always get poor coherence as long as there is water. The authors should rephrase this sentence to reflect this point.

**L229.** This sentence may lead to confusion as it implies water volumes can be calculated using InSAR techniques. Suggest rephrasing to articulate the intended point more clearly.

**L232.** For consistency with the text above, suggest changing to 'affected by tidal and horizontal motion'.

**L237.** Again, I think it'd be really nice to see an estimate of the uplift here, derived from either fringe counting or unwrapping the phase (see also my comments on L198).

**L241.** Amery Ice Shelf.

**L243.** Argument regarding line-of-sight observations only. This is actually only true for Sentinel-1 which, at present, only has one look direction over these ice shelves. Sentinel-1 (or any other sensor for that matter) collected in both ascending and descending orbit could deconvolve those parameters potentially yielding a better impression of subsidence/uplift, or

at the very least a different (and possibly validatory) view of the lake dynamics relative to that gleaned from a single look direction.

Suggest rephrasing the sentence to stress these points, and refocus the sentence away from Sentinel-1 'only' towards a more broad discussion of the different SAR sensors that could possibly be used.

**L244.** I wonder to what extent this sentence is true, since more complicated processing techniques like double-difference InSAR (e.g. Mohajerani et al.; 2021) could presumably help to cancel out ice flow signals. De-tiding observations using a tidal model could also remove vertical motion due to tide (see, for example, MacMillan et al., 2012). Did the authors investigate the applicability of these techniques for improving signal-to-noise over the lake areas? (I'm not suggesting this necessarily needs to be done if not, but I feel a more nuanced/careful discussion of how ice/tide displacement could possibly be mitigated to lake detection easier should be included here).

**L246.** To conclude this section, I think there's big scope to include one or two sentences on the potential advantages of 'next-generation SAR' remote sensing capabilities for lake monitoring. This could involve a discussion of the <6-day imaging capabilities afforded by the launch of Sentinel-1c (~2022), and/or the upcoming (2023) launch of the NASA-ISRO SAR mission (NISAR). While the latter will have a repeat pass time of 12 days over the polar regions, its dual-wavelength (L- and S-band) imaging capabilities may have good potential to circumvent confounding issues such as snow blow and other atmospheric effects, quantify thin/forming ice lid thicknesses etc.

If the authors do not wish add such a discussion, then I recommend editing the title of the study to be sensor specific, e.g. 'The potential of Sentinel-1a/b synthetic aperture radar interferometry for assessing meltwater lake dynamics on Antarctic ice shelves'. (see also my comments regarding the title, L243 and L261-265 above).

**L248.** I think this sentence could (and should) be snapper. Suggest rephrasing to 'This study has provided insight into the utility of InSAR for monitoring meltwater lake dynamics' or similar.

**L261-265.** This is largely repetition of Lines 238-246 which I think can probably be significantly shortened and merged with Lines 266-268. Suggest something like: 'Despite noted limitations to current Sentinel-1 InSAR imaging over parts of Antarctica, we show that InSAR provides promising potential for monitoring meltwater lake dynamics beyond that afforded by conventional, backscatter-only, analyses. Such potential could pave the way for …'.

*Technical Corrections (typological and grammatical errors etc.)*

**Referencing.** I have noticed multiple inconsistencies in the manuscript. Please ensure referencing style is consistent throughout and adheres to The Cryosphere's specific referencing format ([https://www.the-cryosphere.net/submission.html#manuscriptcomposition](https://www.the-cryosphere.net/submission.html#manuscriptcomposition)).

**L5.** Incorrect grammar and sentence structure. Suggest rephrasing to: 'In two case study regions over the Amery and Roi Baudouin ice shelves, East Antarctica, we examine spatial and temporal variations in SAR backscatter intensity and interferometric (InSAR) coherence and phase over several lakes derived from Sentinel-1a/b C-band SAR imagery.

**L15.** Insert commas before and after 'however'.

**L55.** 'By a certain time' is colloquial. Suggest 'by a particular temporal baseline' or similar instead.

**L65.** Remove 'basically' (colloquial usage inappropriate for scientific writing).

**L75-79.** These sentences are repetitive and could easily be merged for conciseness. Also, in the last sentence, I think it's important to explicitly state why you delineated polygons of surrounding snow and ice, as this is unclear.

**L88.** Insert comma after 'For both products'. At end of sentence, also add citation to back up this statement.

**L99.** Pronouns are not to be preceded by 'the', so remove 'the' before Google Earth Engine. (Also true for the likes of 'the Amery Ice Shelf', 'coherence' etc.).

**L102.** Add comma after 'dynamics'. At end of sentence, explicitly state where you perform this analysis (i.e. over the lakes and control (snow/ice) sites). For clarity, this should probably also involve merging the following sentence.

**L105**. Insert comma after 'purpose'. (Note: punctuation errors of this type are a recurring issue and one that I encourage the authors to carefully correct for throughout the manuscript).

**L136.** 'Amery ice shelf' is a pronoun and so should be capitalized. Note that this correction should be carefully applied to all pronouns in the manuscript.

**L150.** Insert commas before and after 'however'.

**L166.** Insert comma after 'gradual'. Regarding the next sentence, I suggest also labelling the circular feature you refer to in the figure, as it took me a while to recognize exactly what you mean.

**L169.** Reference Fig. 5 in the first sentence. The second sentence is also grammatically incorrect and should be edited to state that routine Sentinel-1 coverage commenced in 2017 and to date only acquires data with a repeat-pass of 12 days.

**L172.** Should read '…, with only intermediate sigma-nought values'.

**L214.** Should say 'assess'.

**L215.** Remove 'such as Sentinel-1'.

**L228.** To maintain the flow of the text here, suggest rephrasing this sentence to: 'Beyond coherence, we also demonstrate the potential of interferometric phase for assessing … in areas of high coherence'.

**L231.** Suggest changing 'instant' to 'rapid (sub-weekly) meltwater events', since changes over 6 days can hardly be classified as instant.

**L233.** I think this should say '…an easier detection of stable ice and lake refreezing than coherence and backscatter intensity …'?

**L235.** Incorrect grammar/sentence tense. Suggest rewording to: "While InSAR-based techniques show clear potential for monitoring meltwater lake evolution, there are several key limitations associated with this technique compared with conventional optical- and SAR backscatter-based imaging. First, InSAR requires …'.

**L240.** Replace 'may' with 'can'.

**L241.** 'day' should read 'days'. Also, suggest rewording 'Due to this difference' to 'Due to these differing imaging times' or similar.

**L253.** Sentence beginning 'A generalization'. Reword to 'We show that meltwater detection using backscatter is, however, not straightforward, as meltwater lakes often …'.

**L255.** Replace 'context' with 'circumstance'. Also suggest removing 'i.e. the coherence and interferogram phases'. (this is unneeded technical info for the conclusion).

Above, the authors could also consider rephrasing the text to offer a more well-rounded discussion on the application of SAR in general, rather than specific application of Sentinel-1 data (see my comments regarding title, L243 and L246).

**L256.** 'Besides' should not be used to begin a sentence. Replace with: 'In addition, we show that InSAR-derived information can also be used to observe meltwater lake evolution (and potential drainage) with high accuracy beyond that afforded by conventional backscatter or optical satellite imaging' or similar. Then begin next sentence with: 'Specifically, InSAR coherence information allows for the detection of changes in the …, while interferometric phase can effectively track the spatial and temporal evolution of ice refreezing. Maps of interferometric phase moreover allow for the detection of abrupt lake drainage (or filling) events via changes in the relative displacement of the surface between successive SAR passes'.

**L274.** I think this should say 'WL was responsible … processing and analyzing the results …'.

**L278.** Remove NSF-OPP awards and rest of lines 279 and 280 as these are not relevant to this study.

**L322**. Please cite final (non-TCD) publication.

**Suggested rewriting of introduction (red = suggested rewording).**

Widespread surface meltwater ponding has been observed on Antarctic ice shelves over the past ~X decades (Kingslake et al., 2017). Through seasonal formation and draining of supraglacial lakes, which have the potential to fracture and weaken ice shelves through repeated compression and uplift, respectively (Banwell et al., 2013), such phenomena may have important implications for ice-shelf hydrofracture and collapse (add reference to previous hydrofracture studies here). Therefore, accurately observing the spatial and temporal evolution (filling, drainage or refreezing) of such lakes is pertinent to elucidating the future stability and response of the Antarctic Ice Sheet to climate change.

Given the remote location, widespread area, and harsh climatic conditions in which these lakes form, satellite remote sensing has become the primary method of observing their evolution and dynamics (insert references to back up this claim here). Previous studies have exploited various satellite remote sensing data sources to observe these phenomena; for example, Kingslake et al. (2017) presented an overview of the Antarctic-wide meltwater hydrological network by combining Landsat, WorldView and Aster optical satellite imagery together with historic (pre-satellite) aerial photography. Other work has combined both optical and synthetic aperture radar (hereafter SAR) imagery to detect meltwater features in both Greenland and Antarctica (Benedek and Willis; 2021; Dirscherl et al., 2021), including the detection of subsurface meltwater across East Antarctica's Roi Baudouin Ice Shelf (RBIS; Lenaerts et al., 2016). Such subsurface melting is not detectable from optical-based imagery

alone (#add reference here), emphasizing the potential utility of SAR to better detect total surface meltwater presence.

Despite the potential of optical and SAR imagery in observing surface meltwater, both sensor types have limitations over Antarctica. Polar night and cloud cover, for example, limit data coverage in optical based imagery (Selmes et al., 2013; Williamson et al., 2017), whereas the operating frequencies and active-source configuration of SAR sensors allow for all-weather, day-night imaging (#add reference here). Relative to intuitive representation of meltwater features detected by optical sensors, however, the interpretation of SAR imagery can be complex due to ambiguous backscatter returns and/or image geometry effects (e.g. Fahnestock et al., 1993; Miles et al., 2017; add reference pertaining to geometry effects, as this is another key limitation of SAR interpretation). While cross-polarised (HV or VH) backscatter intensity SAR images generally provide a better contrast between water and ice than single polarization (e.g. HH) images (Miles et al., 2017), such images aren't necessarily always available over Antarctica (#citation possibly of Sentinel-1 acquisition strategy document).

**Papers cited in this review**

McMillan, M. et al. (2012). Mapping ice-shelf flow with interferometric synthetic aperture radar stacking. Journal of Glaciology, doi:10.3189/2012JoG11J072.

Mohajerani, Y. et al. (2021). Automatic delineation of glacier grounding lines in differential interferometric synthetic-aperture radar data using deep learning, Scientific Reports, https://doi.org/10.1038/s41598-021-84309-3.

---

## Referee Comment (RC2)

**The potential of InSAR for assessing meltwater lake dynamics on Antarctic ice shelves**

**Reviewer comments**

**General comments**: This paper evaluated the beneficial of combining SAR amplitude, InSAR coherence and phase information for meltwater lake dynamics. The topic fits well with The Cryosphere journal and it provides useful information for investigation for lake dynamics in Antarctic environment. The selected cases over Amery and Roi Bauouin ice shelves (RBIS) shows that SAR amplitude, InSAR coherence and phase are complimentary for lake dynamics monitoring. However, I think the presented examples may oversimplify the interpretation of SAR amplitude, coherence and phase for monitoring meltwater lake dynamics, as we know other factors, other than seasonal melting-refreeze process, such as weather event (snow, rainfall, et,al) and sensor acquisition geometry (descending/ascending) could also affect amplitude/coherence/phase variation. There are also some other issues with this paper, such as convincing evidence about the lake status in the analysis, and incomplete/confusing data information that were used in the study. Please see my comments for details:

**Specific comments**

1) Instead of just few selected data, please provide a complete time series amplitude, coherence and phase analysis for the cases in Fig 4 & Fig 5. I think this would still show the benefits of different information (amplitude, coherence and phase), but it would provide a more objective sense/perspective for reader to understand potential drawbacks of each different information. Incomplete data also make some of the statements confusing in the paper. For example, Line 155-157, it talked about amplitude/coherence for summer melting, but there are no SAR data shown in the Fig 4.

2) Please provide evidence when refer to melting/refreeze/frozen status of the lake to make your statement convincing. For example, the authors explained the decorrelation in Jan 2017 data is due to melting (Line 195-196) in Fig 7. However, in this same figure, we see the Jan 2018 shows very good coherence and phase pattern. I would assume the area would be in similar freeze/melt status at approximate same time of different years. I am not sure whether the low coherence in Jan 2017 is due to melting or maybe other weather events. I think it would be helpful to collect some other information, such as temperature information from other sources, to support your statement. For all other data analysis, if it's possible to collect some external information such as temperature or optical imagery, I would suggest doing so that it's more convincing when you state its under melt or refreeze or frozen status.

3) Incomplete data information. For the time series of mean and standard deviation over selected polygons mentioned in Fig 2, How many SAR data are used for this calculation and what are their acquisition times? are the mean and std for all the polygons shown in Fig 1? It might be helpful to provide complete data list in text or supplement. Are the coherence data for Amery all 12-days product? It would be not meaning to mix 6-days or 12 days data together to analyze lake-related information, as temporal difference would change that a lot.

Please show the outline of the sentinel-1 data in the last panel of Fig 1. What is the data coverage used in this study? Table 1 shows RBIS SLC data is from 2017/7/25—2018/4/15, however, in fig 2, the data coverage for RBIS is from 2017/1-2018/1, it is so confusing. Please provide accurate

info for data you used. Also, the GRD and SLC time coverage are different as shown in fig 2, not sure how does this happen. I would assume you need to analyze amplitude/coherence/phase comparison for all data.

4) Are there any different characteristics in amplitude/coherence/phase between supraglacial and englacial lake?

**Technical correction:**

Line 113, 'the results', please be more specific.

Fig 2. Please provide complete legends for subplots 2-4. Fig caption are not complete. It only takes about the time series, but not the specific amery a, b, d and RBIS examples.

---

## Author Comment (AC1)

Response to Referee 1 on tc-2021-169

First, we would like to thank the Referee for reviewing and commenting the manuscript, which improves the quality of the manuscript. Please find the item-by-item reply below, with the original comments in *italics* and the responses in blue. All the suggested changes will be implemented in the revised text that will be uploaded.

***General Comments***
*My main concern pertains to the overall readability of the manuscript. While it is clear that the authors understand the background literature, methods, key results and implications of their research, their writing style is in general unconventional for a piece of scientific writing, insofar as it is highly verbose, often grammatically and/or typographically incorrect and, hence, difficult to follow/comprehend from a reader's point of view. This is further confounded by what appears to be inconsistencies in the clarity/style of writing adopted in different sections of the manuscript, missing information needed to fully understand the datasets and/or logic of arguments presented, and the occasional lack of relevant citations throughout the text (see my specific and technical comments for further details).*
*To address these issues, I recommend that all coauthors take the time to carefully restructure the wording of the manuscript to: a) more logically explain (and justify fully) the choice of all techniques and methodological decisions used/taken, b) correct typographical/grammatical errors and, c), cut down and hence improve the overall focus/narrative of the text. To assist the authors in this regard, I have made some suggestions on how the first two paragraphs of the introduction could be rewritten (see bottom of this document). If needed, the authors may also find the following resource (https://www.the-cryosphere.net/submission.html#english) and links therein helpful.*

We appreciate the suggestions and the text will be modified according to the specific comments.

***Title.*** *'InSAR' is an abbreviation and hence inappropriate for use in a title. Suggest rephrasing title to: 'The potential of synthetic aperture radar interferometry for assessing meltwater lake dynamics on Antarctic ice shelves' or similar instead. (see also my comments regarding Line 246).*

This will be changed in the revised text.

***Line 2.*** *Suggest replacing 'Yet' with 'Despite these phenomena'. Replace 'or' with 'and'.*

This will be implemented in the revised text. As language corrections can be directly implemented in the script, all comments regarding rephrasing/grammatical corrections will be replied as 'done' in the following text of this reply.

***L3.*** *Suggest either ending sentence after 'limited', or briefly explaining what the limitations of optical satellite imagery are here.*

Added 'during polar night and in cloudy conditions' after 'satellite imagery'.

*L8. Change 'The analysis' to 'Our analysis'. At end of this sentence, change 'confounded' to either 'hard to distinguish' or 'indistinguishable'. Then change next sentence to: 'Despite this finding, we show using a combination of backscatter and InSAR observations that lake dynamics can be effectively captured during other non-summertime months'.*

Done.

*L11. Sentence beginning 'In particular'. For conciseness, suggest merging this and next sentence to: 'Moreover, our findings highlight the utility of InSAR-based observations for discriminating between refrozen ice and subsurface meltwater, and indicate the potential for phase-based detection and monitoring of rapid meltwater drainage events'.*

Done.

*L56. Remove 'The' and begin sentence 'Coherence is considered an indicator of changes'. Note also that phase difference does not correspond to the 'average' difference, but a very precise measurement of whole wavelength (and some fractional component) range difference. Suggest editing the rest of the sentence to reflect this.*

Done.

*L59. Sentence beginning 'This combination'. Who expects this? Add a reference to back up this claim, otherwise say 'We expect this' or similar.*

Will be changed to 'We expect this combination…'

*L67. Suggest changing to: "However, the value added using InSAR for such applications has not yet been examined".*

Done.

*L68-71. The structure of this paragraph is rather difficult to follow. Suggest beginning with "In this paper, we assess the potential of C-band backscatter and InSAR data to ... For this purpose, we use a combination of backscatter, coherence and phase information to monitor recent meltwater features over two East Antarctic locartions – the Amery and Roi Baudouin (RBIS) ice shelves – using data collected by Sentinel-1a/b in 2017/2018. To supplement the interpretation of our (In)SAR-based analyses, we also utilize spatially and temporally collocated optical satellite data.*

The whole paragraph will be changed as suggested.

*L75. Sentence is very long and could be split in two after reference to Lenaerts et al. (2016).*

Done.

*L81. Why was a lake dataset not available? The reader shouldn't need to guess this, so state explicitly here. (I think you mean that no previously published dataset exists?). When mentioning Landsat data, also point readers to Section 2.2 for reference.*

Will be changed from 'a reference lake data set was not available' to 'no previously published dataset with in-situ study is available'. Reference to Section 2.2 is added.

*L83. Sentence beginning 'Our lake class'. If my understanding of the above is correct, then this sentence is confusing as it suggests that a preexisting lake dataset does indeed exist. If so, why didn't you use that here? Please either edit sentence of clarify or remove from the text.*

Previous studies observed the Amery Ice Shelf with satellite imagery, but no in-situ dataset is available. We will implement a modification in Line 81 to clarify this.

*Figure 1. This is a nice figure, but please add latitude/longitude information to each panel so that readers can easily deduce locations. Please also consider showing two additional, zoomed out panels showing the location of each subset within both ice shelves, and (if necessary) in these and all pre-existing panels, add the ice sheet grounding line for reference (e.g. https://doi.org/10.7280/D1VD6G). (I know that the lower right panel shows this to a certain extent, but it's difficult to see details. In general, I also find the labels rather small and difficult to locate, so these could also be enlarged (plenty of space on figures to do this).*

The suggestions will be implemented.

*Figure 1 caption. Please define RBIS in full in caption. Replace 'close ups' with 'inset' or 'detail'. Insert comma after 'panels'. Change 'delineated in black curves' to 'delineated as black curves'. In next sentence, what does 'indices' refer to? Labels? If so use 'labels' instead for clarity. Suggest rephrasing following sentence to read: "...are also delineated for comparison against backscatter intensity and coherence values observed over lakes (Fig. 2)' or similar. For general readability, next sentences could/should read: 'Panel R2 illustrates the lake feature shown in Figure [insert number here]. Inset shows location of the analyzed locations'. Please also state in caption which band/band combinations are shown (Landsat).*

Done.

*Table 1 caption. Should read '... used in this study'.*

Done.

*L99. Please add more information (and references if necessary) on how the images were denoised, calibrated and corrected here. (The reader shouldn't have to look it up for themselves).*

Information on calibrated top-of-atmosphere (TOA) reflectance (Chander, 2009) Landsat images will be added in the revised script.

*L92.* I presume this is a typo and should say 10x10 m resolution (i.e. the native resolution of IW GRDH; https://sentinels.copernicus.eu/web/sentinel/technical-guides/sentinel-1-sar/products-algorithms/level-1-algorithms/ground-range-detected/iw)? If not, then how does this impact the rest of your backscatter analyses?

*Also, if my understanding is correct then NRCS is identical to radiometrically calibrated, sigma-nought ($\sigma0$) backscatter imagery. Sigma-nought backscatter is the much more commonly adopted term in the geosciences (RCS more so in engineering), so given the likely readership of The Cryosphere, I would instead refer to $\sigma0$ in place of NCRS here and universally throughout the manuscript. Following this point, be careful that the imagery you downloaded from Google Earth Engine isn't already in sigma-nought format, as the 'radiometric calibration' you mention GEE perform above may imply.*

About the resolution, the Single Look Complex images of Sentinel-1 have a 20 m (azimuth) x 4.5 m (ground range). The GRDH are derived by averaging around 4.4 looks in range to make the resolution approximately 20 x 20 m. The products are then provided in an upsampled format, with pixel spacing/posting of 10 m. So we believe that the value of 20 m reported in the manuscript (that is the relevant one for the equivalent number of looks, or else for the radiometric precision when averaging areas of distributed targets such as ice, snow and water) is correct.

We appreciate the advice of using a consistent nomenclature throughout the manuscript. In the revised version 'NRCS' will be therefore changed into '$\sigma0$', including the figures. About the calibration, our understanding (see https://developers.google.com/earth-engine/datasets/catalog/COPERNICUS_S1_GRD) is that GEE already provides radiometrically calibrated GRD images, and, as such, in $\sigma0$ format.

*L95.* Confusing. If you have multi-looked the SLC to create your own GRD/Sigma-nought imagery then why did you bother downloading GRD imagery from GEE? Clarify here.

The main difference between our products and the imagery downloaded from GEE is that our analysis on the RBIS focuses on ascending track 59, whose data were not available on the rolling archive of Sentinel Scihub before July 25, 2017. To analyse the lake formation since 2016 (shown in Figure 3), we conveniently made use of the imagery available on GEE. This will be clarified in the revised manuscript.

*L97.* Why is a geoid only used for Amery? I expect this is a typo?

At the moment of processing we lacked a detailed DEM for Amery and therefore used the Geoid as this closely resembles the ice shelf surface. Although we agree that this could be improved (e.g. by using REMA), we expect it will not change any of the conclusions of the analyses. This is proven in Fig. 4 and Fig. 7, as the meltwater features in the InSAR images are in the same location as in Landsat images.

*L98+ This is a good example of missing methods I mentioned in my general comments. Here, I am surprised to see absolutely no information about how the authors generated their interferograms. This must be added, including, as a minimum, information on e.g.*

*temporal/perpendicular baselines used, type of processing performed (I assume single-pass DInSAR?) and (if used) any DEM used to remove topographical phase.*

The overview of DORIS will be added in the revised manuscript.

*L99-100. Again, for clarity/reproducibility purposes, much more info should be included here on e.g. the band/band combinations used, pixel resolutions of the data, and any relevant pre-processing steps you applied to your Landsat imagery. Stating that you simply downloaded them off GEE is not appropriate for a scientific paper.*

TOA Tier 1 Landsat surface reflectance data (Chander) of bands (RGB) were downloaded from GEE at their native 30 m pixel resulution without any additional pre-processing steps. This will be added in the revised manuscript.

*L105. I think it's important to explicitly state here that for each class type analyzed, you calculated the mean and standard deviation backscatter for all observed features. As written this is not obvious, and leads to confusion over what the difference is between 'lake class' and the 'individual lakes' (for this reader at least ...). This could also be made more explicit in the caption of Figure 2.*

The sentence will be completed as 'For this purpose, the temporal variations in $\sigma 0$ are first compared per lake, snow, ice class by analysing the time series of the mean and standard deviation backscatter for each class (i.e. lakes, snow and ice) and for each individual lake.' Also added 'Mean and standard deviation are calculated for all features presented.' to the caption.

*L106. For clarity, 'mono-dimensional' should be changed to 'cross-sectional'.*

Done.

*L107. Repetition of coherence, NCRS and phase. Suggest restructuring sentence to avoid this.*

Will be changed from '…combining NRCS, coherence and phase information...' to '...combining SAR backscatter intensity with InSAR information...'

*L112. Suggest changing to: 'The mean sigma-nought timeseries of lakes, snow and ice (cf. Section 2.2) display strong seasonal variability, consistent with the changing nature of both surface snow and ice properties and the evolution of supraglacial lakes through time (Figure 2). On Amery Ice Shelf, our observations reveal...'.*

Done.

*L113. This is written in an odd manner which implies that snow transforms into lakes and then ice, which is not what the authors intend to say. What I think they mean to say is that snow, lakes and ice for the most part display different (though reasonably constant) backscatter*

*properties throughout the year, with the exception of JF when the backscatter associated with snow and lakes fall rapidly.*

We understand the confusion. Now the sentence has been changed from 'The results on Amery ice shelf show that the NRCS decreases from snow (˜0 dB) to lakes (˜-5 dB) and ice (˜-10 dB) during fall, winter, spring…' to 'The results on Amery ice shelf show that the $\sigma 0$ has different levels for snow ~0 dB), lakes (~-5 dB) and ice (~-10 dB) and is relatively constant (fluctuations within ~1 dB)…'.

*L116-125. To shorten the text here, I question whether the authors even need to discuss (and, in Figure 2, show) the individual lakes because for the most part, the average of multiple mapped lakes makes seems to support their arguments just as well. In this regard the individual lake observations are a slight distraction from the overall story revealed by the class averages, so I think they could probably be removed. (as the authors show, there is significant variability from one lake to the next, so focusing in on specific lakes only serves to deviate from what's happening on the whole). This is also largely true for the coherence discussion of RBIS a and f in Section 3.2, as your later coherence images (Figure 7) in any case demonstrate the process of refreezing in a much more convincing way).*

We agree, and the figure and the texts will be modified.

*L126-134. For clarity, suggest editing sentence to say '...of select cross-sectional transects. In the case of both RBIS 'a' and Amery 'd' (location shown in Figure 1), for example, backscatter timeseries show significant inter-annual variation (Figure 3)'.*

Done.

*L129. 'Border of low NRCS and inner areas of high NRCS'. Revise this sentence to explicitly state that this refers to the edge and central regions of the lake, respectively.*

Will be changed from 'After this, a clear spatial pattern emerges with borders of low NRCS and inner areas of high NRCS, followed again by a new area wide decrease in NRCS in the Dec. 2017 - Jan. 2018 melting season.' to 'Subsequently, a clear spatial pattern emerges with borders of low $\sigma 0$ at the edges and high $\sigma 0$ in the central regions, which respectively refer to the edge and central regions of the lake. This pattern is followed again by a new area-wide decrease in $\sigma 0$ in the Dec. 2017 - Jan. 2018 melting season.'

*L136-152. These are clear, well written paragraphs. They are logical and concise, and could be considered a model for how the rest of the manuscript should be written.*

We appreciate the suggestion.

*Figure 2. See my comments regarding L116-125 above. If the authors choose to retain the analysis of individual lake features, then they should make the lines thicker as these are currently very difficult to see both on-screen and in print. To enhance visibility of these lines, I'd also consider making the standard deviation ribbons more transparent as these currently*

*dominate/clutter the figure. As per Figure 1, I also think the labels (and legend especially) should be made bigger/more prominent.*

*For the coherence plots (and to a lesser degree sigma-nought), I wonder if the high frequency variability discussed by the authors could be smoothed out using something like a running mean? While this variability is interesting, it's a little distracting, and is later largely ignored in the text anyway.*

We will fix the figure in the revised manuscript.

**Figure 2 caption.** *Sentence should read '... over the Amery and Roi Baudouin ice shelves (see Fig. 1 for locations). Change 'Moments ...' with 'Times with a lack of 6/12-day...'.*

Done.

**Figure 3.** *As per Figs 1 and 3, please make all labels larger. Please also add lat/longs to both maps along with scalebars. Also consider zooming both images to show more detail over lakes (bottom right panel especially).*

This will be implemented to the revised script.

**Figure 4.** *Nice figure! As above regarding label size.*

Thanks for the suggestion.

**Figure 4 caption.** *Remove 'synoptic' (incorrect usage in this context) in first sentence, and cross-reference Fig. 1 for location at end. In next sentence, remove 'the' proceeding coherence, and add '... and resulting phase difference interferograms are shown ...'. In next sentence, should say 'The high frequency fringes surrounding each lake represent a convolution of both ice flow and tidal motion'. In the last sentence, please state which band/band combinations are shown (Landsat).*

Done. Thanks for the correction.

**Figure 5.** *Really nice figure, but please add lat/longs and scalebars.*

They will be added.

**Figure 5 caption.** *Unnecessary use of 'right' which should be removed. In the next sentence, change 'hereby' to 'hereafter'. In the following sentence, change 'reported' to 'shown (right panels)'. Please also state which band/band combinations are shown.*

Done.

**L147.** *I think this should say 'between Oct. 2017 and Jan. 2018'. Change 'polygons' to 'surveyed snow, ice and lake areas'.*

This will be corrected.

*L154. Amery Ice Shelf.*

Done.

*L159. Replace 'brighter' with 'greater'.*

Done.

*L172. Why 'possibly'? Provide evidence to justify claims here. (Also, RBIS is a rather slow flowing ice shelf, so horizontal displacement should not influence phase coherence over 12 days as much as one might think (see Mohajerani et al. (2021) who were able to map GLs across this region Antarctica with good coverage using double difference InSAR. This technique requires almost perfect coherence, suggesting 12 days is more than sufficient here).*

*In the next sentence (beginning 'In Oct. 17...'), I think better referencing to Figures 2 and 5 is needed as I don't see any change in coherence from Figure 5 alone.*

Our interpretation is that our region of interest is more affected by weather condition especially in winter, so the coherence from the 12-day revisit is corrupted mainly by wind/precipitation. We plan to add information from ERA5 or equivalent weather data to assist the understanding.

Reference to the figures is added.

*L176. Change tense to be consistent with the rest of the paragraph. Also, while what you go on to say in Lines 176-177 is technically true, visually I can't tell the difference between the lakes you are discussing and the drainage network. Suggest rewriting this sentence for clarity to specifically emphasize the observed change from a lake to a (presumably) connected drainage network through time.*

*This is a really nice observation by the way, demonstrating in a compelling manner the utility of coherence to see what simple optical and/or backscatter images cannot.*

The tense will be corrected. Will be changed from '…which is not straightforward to see in the NRCS or optical imagery. This highlights the increased potential for coherence over the backscatter intensity in delineating the lake network.' to '...which is shown as dark strips between the highlighted lakes in the lower middle panel of Fig. 5. The patterns are clearly newly formed compared to the lower left panel of Fig. 5. This change is not straightforward to see in the $\sigma 0$ or optical imagery.'

*L179. Suggest rewriting to begin: 'Interferometric phase difference maps (Figure 4) emphasize... Amery ice Shelf.*

Done.

*L180.* Initially I didn't see any fringes you refer to (c/w Amery c for example) given the dominance of the high frequency (ice flow) fringes surrounding the lakes, but then I realized you meant the very low frequency fringes on the lakes themselves (~1 cycle of -π to π only). I suggest you state this more clearly (and perhaps label the figure accordingly) so that readers don't incorrectly focus in on the high frequency fringes.

The labels will be improved and the low-frequency patterns will be specified in the revised manuscript.

*L182.* Edges of what? I can work out what you mean, but this can be written more clearly for ease of reading. Possibly also consider citing appropriate figures and panels.

Edges of lake Amery b. This will be added to the sentence.

*L183.* Suggest writing as '...... increase through time indicates the presence of lakes until October 2017, followed... in November of that year. Consistent with our InSAR-based observations Landsat ...'.

Done.

*L179-203.* In general, this is another clearly written and easy to comprehend series of paragraphs compared with the earlier section of the manuscript.

Thank you for the suggestion.

*L179-186.* Regarding Amery Ice Shelf, what (if anything) can we learn about the detection of the hydrological network that is clearly visible in Figure 4 (top row), and which disappears after March 2017 in the coherence images? (suggesting formation between Mar 11 and 17th and persistent presence (freezing?) thereafter). This is a visually striking feature in the center of these panels that I was surprised to see no discussion of here and/or in Section 3.2.

We appreciate the suggestion, and the discussion will be added in the revised manuscript.

*L187.* I'm not quite sure I follow this, as the color scheme always goes from blue to blue. Suggested rewriting for clarity.

In the lower left panel, the patterns are shown (from left to right) as blue - green - yellow - red in the left part of the highlighted circular area, blue - red - yellow - green in the right part of this circular area, and further right blue - green - yellow - red outside the highlighted circle. This generally forms a whirl-like feature. We will try to clarify that in the text.

*L198.* How big was this uplift? I think that would be a valuable addition here, and can be estimated either through unwrapping the phase or counting the fringes.

We agree that it would indeed be very interesting. By counting the fringes, the redult is approximately 7 fringes, each measuring 2.8 cm in the line-of-sight. Assuming a vertical movement, this corresponds to an uplift of approximately 24 cm (taking into account an

incidence angle of approximately 35°). However, without data for validation, we were cautious in providing an exact number.

*L190. And presumably some tidal component, as \*I think\* tides haven't been removed? (see also my comments on the omission of any methods detailing exact InSAR processing above).*

We agree. This will be added to the revised script.

*L194-203. Great series of observations.*

Thank you.

*Figure 6. Nice figure, but please add lat/longs and scalebars to all panels.*

This will be implemented.

*Figure 6 caption. Replace 'interferometric phases' with 'interferometric phase'. For brevity, suggest rewording next sentence as 'Two near-contemporaneous Landsat 8 panchromatic (band 8) images are also shown (right panels)'.*

Done.

*L201. Replace 'starting at the edges' with 'towards the center of the lake' or similar.*

Done.

*Figure 7. Very nice series of observations! Enlarge labels and add lat/longs and scalebar, though.*

This will be implemented.

*Figure 7 caption. Replace 'interferogram phases' with 'interferograms' or similar. Please also state which band/band combinations are shown for Landsat imagery.*

Done.

*L204. I'm not sure I completely follow what you're trying to say here, as the sentence contains a grammatical error. Suggest rephrasing for greater clarity.*

Will be changed from 'The interferogram shows similar results here, but with the added value that the interpretation of high-low backscatter compared to the surroundings is less ambiguous.' to 'The interferogram shows similar results here. However, it is still ambiguous to interpret the higher/lower backscatter intensity of the lakes compared to the surroundings. Adding interferograms to the observation helps to reduce this ambiguity.'

*L205. Suggest beginning this section like: 'Using SAR-based observations acquired across two East Antarctic ice shelves, we present evidence of the utility of backscatter ...'.*

Will be changed into 'Using SAR-based observations acquired across two East Antarctic ice shelves, this study presents evidence of the utility of backscatter intensity and coherence to assess meltwater lake dynamics.'

*L213. Change 'Coherence' to 'Interferometric coherence'.*

Done.

*L215. And all other types of SAR SLC data ... not just that acquired by Sentinel-1.*

Sentinel-1 will be removed from the sentence.

*L222. I think its important to stress here that low coherence isn't just about refreezing (or not). Radar waves are fully attenuated by water, so you will always get poor coherence as long as there is water. The authors should rephrase this sentence to reflect this point.*

This will be changed from '…while meltwater lakes show a low coherence due to the constantly changing ice/water interface', to '…while meltwater lakes show a low coherence due to the constantly changing ice/water interface and the increased attenuation due to the presence of water'.

*L229. This sentence may lead to confusion as it implies water volumes can be calculated using InSAR techniques. Suggest rephrasing to articulate the intended point more clearly.*

We intended to say that estimating the water volumes is not within the scope of this study. To avoid confusion, this sentence will be removed.

*L232. For consistency with the text above, suggest changing to 'affected by tidal and horizontal motion'.*

Done.

*L237. Again, I think it'd be really nice to see an estimate of the uplift here, derived from either fringe counting or unwrapping the phase (see also my comments on L198).*

The reply is the same as for L198.

*L241. Amery Ice Shelf.*

Done.

*L243. Argument regarding line-of-sight observations only. This is actually only true for Sentinel-1 which, at present, only has one look direction over these ice shelves. Sentinel-1 (or any other sensor for that matter) collected in both ascending and descending orbit could deconvolve those parameters potentially yielding a better impression of subsidence/uplift, or*

*at the very least a different (and possibly validatory) view of the lake dynamics relative to that gleaned from a single look direction.*

*Suggest rephrasing the sentence to stress these points, and refocus the sentence away from Sentinel-1 'only' towards a more broad discussion of the different SAR sensors that could possibly be used.*

We will add 'With SAR acquisitions from sensors in both ascending and descending orbits, it is however possible to better quantify the lake subsidence/uplift.' to the paragraph.

*L244. I wonder to what extent this sentence is true, since more complicated processing techniques like double-difference InSAR (e.g. Mohajerani et al.; 2021) could presumably help to cancel out ice flow signals. De-tiding observations using a tidal model could also remove vertical motion due to tide (see, for example, MacMillan et al., 2012). Did the authors investigate the applicability of these techniques for improving signal-to-noise over the lake areas? (I'm not suggesting this necessarily needs to be done if not, but I feel a more nuanced/careful discussion of how ice/tide displacement could possibly be mitigated to lake detection easier should be included here).*

This discussion will be included.

*L246. To conclude this section, I think there's big scope to include one or two sentences on the potential advantages of 'next-generation SAR' remote sensing capabilities for lake monitoring. This could involve a discussion of the <6-day imaging capabilities afforded by the launch of Sentinel-1c (~2022), and/or the upcoming (2023) launch of the NASA-ISRO SAR mission (NISAR). While the latter will have a repeat pass time of 12 days over the polar regions, its dual-wavelength (L- and S-band) imaging capabilities may have good potential to circumvent confounding issues such as snow blow and other atmospheric effects, quantify thin/forming ice lid thicknesses etc.*

*If the authors do not wish add such a discussion, then I recommend editing the title of the study to be sensor specific, e.g. 'The potential of Sentinel-1a/b synthetic aperture radar interferometry for assessing meltwater lake dynamics on Antarctic ice shelves'. (see also my comments regarding the title, L243 and L261-265 above).*

This will be added to the discussion.

*L248. I think this sentence could (and should) be snapper. Suggest rephrasing to 'This study has provided insight into the utility of InSAR for monitoring meltwater lake dynamics' or similar.*

Done.

*L261-265. This is largely repetition of Lines 238-246 which I think can probably be significantly shortened and merged with Lines 266-268. Suggest something like: 'Despite noted limitations to current Sentinel-1 InSAR imaging over parts of Antarctica, we show that InSAR provides*

*promising potential for monitoring meltwater lake dynamics beyond that afforded by conventional, backscatter-only, analyses. Such potential could pave the way for ...'.*

Done.

**Referencing.** *I have noticed multiple inconsistencies in the manuscript. Please ensure referencing style is consistent throughout and adheres to The Cryosphere's specific referencing format (https://www.the-cryosphere.net/submission.html#manuscriptcomposition).*

Thanks for the suggestion. This will be double-checked in the revised script.

**L5.** *Incorrect grammar and sentence structure. Suggest rephrasing to: 'In two case study regions over the Amery and Roi Baudouin ice shelves, East Antarctica, we examine spatial and temporal variations in SAR backscatter intensity and interferometric (InSAR) coherence and phase over several lakes derived from Sentinel-1a/b C-band SAR imagery.*

Done.

**L15.** *Insert commas before and after 'however'.*

We agree that we dropped a lot of commas within sentences. This will be thoroughly checked.

**L55.** *'By a certain time' is colloquial. Suggest 'by a particular temporal baseline' or similar instead.*

Done.

**L65.** *Remove 'basically' (colloquial usage inappropriate for scientific writing).*

Done.

**L75-79.** *These sentences are repetitive and could easily be merged for conciseness. Also, in the last sentence, I think it's important to explicitly state why you delineated polygons of surrounding snow and ice, as this is unclear.*

We will add '…manually delineated sample polygons of snow and ice surfaces based on Landsat imagery for studying the difference between meltwater lakes and the solid surrounding regions'.

**L88.** *Insert comma after 'For both products'. At end of sentence, also add citation to back up this statement.*

Done. This statement (that only HH-polarisation is available) comes from our random searching over several locations in Antarctica. For reliability, we will change 'Antarctica' into 'the studied ice shelves'.

*L99. Pronouns are not to be preceded by 'the', so remove 'the' before Google Earth Engine. (Also true for the likes of 'the Amery Ice Shelf', 'coherence' etc.).*

Thanks for the correction. This will also be checked.

*L102. Add comma after 'dynamics'. At end of sentence, explicitly state where you perform this analysis (i.e. over the lakes and control (snow/ice) sites). For clarity, this should probably also involve merging the following sentence.*

Done.

*L105. Insert comma after 'purpose'. (Note: punctuation errors of this type are a recurring issue and one that I encourage the authors to carefully correct for throughout the manuscript).*

Done.

*L136. 'Amery ice shelf' is a pronoun and so should be capitalized. Note that this correction should be carefully applied to all pronouns in the manuscript.*

This will be corrected.

*L150. Insert commas before and after 'however'.*

Done.

*L166. Insert comma after 'gradual'. Regarding the next sentence, I suggest also labelling the circular feature you refer to in the figure, as it took me a while to recognize exactly what you mean.*

Done. An arrow to the feature will be added.

*L169. Reference Fig. 5 in the first sentence. The second sentence is also grammatically incorrect and should be edited to state that routine Sentinel-1 coverage commenced in 2017 and to date only acquires data with a repeat-pass of 12 days.*

Reference will be added. Changed sentence from 'Since the Sentinel-1 SLC temporal coverage is lower than for Amery, SLC coverage only started in July 2017 (Fig. 5).' to 'Differently from data on Amery Ice Shelf, the Sentinel-1 SLC acquisition only started in July 2017, with a 12-day revisit (Fig. 5).'

*L172. Should read '…, with only intermediate sigma-nought values'.*

Done.

*L214. Should say 'assess'.*

Done.

*L215. Remove 'such as Sentinel-1'.*

Done.

*L228. To maintain the flow of the text here, suggest rephrasing this sentence to: 'Beyond coherence, we also demonstrate the potential of interferometric phase for assessing ... in areas of high coherence'.*

Done.

*L231. Suggest changing 'instant' to 'rapid (sub-weekly) meltwater events', since changes over 6 days can hardly be classified as instant.*

Done.

*L233. I think this should say '...an easier detection of stable ice and lake refreezing than coherence and backscatter intensity ...'?*

Yes. That was an absent-minded mistake. It will be corrected.

*L235. Incorrect grammar/sentence tense. Suggest rewording to: "While InSAR-based techniques show clear potential for monitoring meltwater lake evolution, there are several key limitations associated with this technique compared with conventional optical- and SAR backscatter-based imaging. First, InSAR requires ...'.*

Done.

*L240. Replace 'may' with 'can'.*

Done.

*L241. 'day' should read 'days'. Also, suggest rewording 'Due to this difference' to 'Due to these differing imaging times' or similar.*

Done.

*L253. Sentence beginning 'A generalization'. Reword to 'We show that meltwater detection using backscatter is, however, not straightforward, as meltwater lakes often ...'.*

Done.

*L255. Replace 'context' with 'circumstance'. Also suggest removing 'i.e. the coherence and interferogram phases'. (this is unneeded technical info for the conclusion).*

*Above, the authors could also consider rephrasing the text to offer a more well-rounded discussion on the application of SAR in general, rather than specific application of Sentinel-1 data (see my comments regarding title, L243 and L246).*

Done.

**L256.** *'Besides' should not be used to begin a sentence. Replace with: 'In addition, we show that InSAR-derived information can also be used to observe meltwater lake evolution (and potential drainage) with high accuracy beyond that afforded by conventional backscatter or optical satellite imaging' or similar. Then begin next sentence with: 'Specifically, InSAR coherence information allows for the detection of changes in the ..., while interferometric phase can effectively track the spatial and temporal evolution of ice refreezing. Maps of interferometric phase moreover allow for the detection of abrupt lake drainage (or filling) events via changes in the relative displacement of the surface between successive SAR passes'.*

Done.

**L274.** *I think this should say 'WL was responsible ... processing and analyzing the results ...'.*

Done.

**L278.** *Remove NSF-OPP awards and rest of lines 279 and 280 as these are not relevant to this study.*

That was the standard citation format required by the publisher (please refer to the PGC acknowledgement site https://www.pgc.umn.edu/guides/user-services/acknowledgement-policy/).

**L322.** *Please cite final (non-TCD) publication.*

This paper gives the following information: Review status: this preprint was under review for the journal TC. A revision for further review has not been submitted.

**References**
Chander, G., Markham, B. L., and Helder, D. L.: Summary of current radiometric calibration coefficients for Landsat MSS, TM, ETM+, and EO-1 ALI sensors, Remote Sensing of Environment, 113, 893–903, https://doi.org/https://doi.org/10.1016/j.rse.2009.01.007, 2009.

---

## Author Comment (AC2)

Response to Referee 2 on tc-2021-169

Thank you for reviewing and commenting the manuscript. Please find the item-by-item reply below, with the original comments in *italics* and the responses in blue.

*General comments: This paper evaluated the beneficial of combining SAR amplitude, InSAR coherence and phase information for meltwater lake dynamics. The topic fits well with The Cryosphere journal and it provides useful information for investigation for lake dynamics in Antarctic environment. The selected cases over Amery and Roi Bauouin ice shelves (RBIS) shows that SAR amplitude, InSAR coherence and phase are complimentary for lake dynamics monitoring. However, I think the presented examples may oversimplify the interpretation of SAR amplitude, coherence and phase for monitoring meltwater lake dynamics, as we know other factors, other than seasonal melting-refreeze process, such as weather event (snow, rainfall, et,al) and sensor acquisition geometry (descending/ascending) could also affect amplitude/coherence/phase variation. There are also some other issues with this paper, such as convincing evidence about the lake status in the analysis, and incomplete/confusing data information that were used in the study.*

We would like to thank Referee 2 for the positive comments on the potential. We agree with the arguments that other factors such as changes in snow properties and sensor effects also may impact the signals. This was in fact mentioned already in the manusript, but it was perhaps not given the proper weight. In the revised version, we will stress the role of these other interpretations more clearly to avoid the impression of oversimplification.

*Specific comments*
*1) Instead of just few selected data, please provide a complete time series amplitude, coherence and phase analysis for the cases in Fig 4 & Fig 5. I think this would still show the benefits of different information (amplitude, coherence and phase), but it would provide a more objective sense/perspective for reader to understand potential drawbacks of each different information. Incomplete data also make some of the statements confusing in the paper. For example, Line 155-157, it talked about amplitude/coherence for summer melting, but there are no SAR data shown in the Fig 4.*

Starting from lines 155-157, we perhaps caused confusion by mentioning the melt of the ice. The lines refer to the background blue ice area, rather than the lakes. And the blue ice features are shown in the second panel at the bottom of Fig. 4 (RGB bands of Landsat image). We will try to clarify it in the revised manuscript.

As for the complete time series, it has 60 days of acquisition, and showing all the NRCS, coherence and phase images will result in 3*60=180 images. This is not ideal to show in one figure, but if it is required and if the editor agrees, we could add it as Supplementary Material.

*2) Please provide evidence when refer to melting/refreeze/frozen status of the lake to make your statement convincing. For example, the authors explained the decorrelation in Jan 2017 data is due to melting (Line 195-196) in Fig 7. However, in this same figure, we see the Jan 2018 shows very good coherence and phase pattern. I would assume the area would be in similar freeze/melt status at approximate same time of different years. I am not sure whether*

*the low coherence in Jan 2017 is due to melting or maybe other weather events. I think it would be helpful to collect some other information, such as temperature information from other sources, to support your statement. For all other data analysis, if it's possible to collect some external information such as temperature or optical imagery, I would suggest doing so that it's more convincing when you state its under melt or refreeze or frozen status.*

Based on previous studies, we are sure that it is melting, due to the backscatter drop that would also be identified as melting in a typical melt-detection algorithm. We appreciate the referee's suggestion in using other external data sources. It is difficult, however, to acquire completely concurrent Landsat images due to the cloud cover. We will therefore include ERA-5 data or similar climate products in the revised manuscript to show it more clearly.

*3) Incomplete data information. For the time series of mean and standard deviation over selected polygons mentioned in Fig 2, How many SAR data are used for this calculation and what are their acquisition times? are the mean and std for all the polygons shown in Fig 1? It might be helpful to provide complete data list in text or supplement. Are the coherence data for Amery all 12-days product? It would be not meaning to mix 6-days or 12 days data together to analyze lake-related information, as temporal difference would change that a lot.*
*Please show the outline of the sentinel-1 data in the last panel of Fig 1. What is the data coverage used in this study? Table 1 shows RBIS SLC data is from 2017/7/25—2018/4/15, however, in fig 2, the data coverage for RBIS is from 2017/1-2018/1, it is so confusing. Please provide accurate info for data you used. Also, the GRD and SLC time coverage are different as shown in fig 2, not sure how does this happen. I would assume you need to analyze amplitude/coherence/phase comparison for all data.*

We are sorry to hear that this information seems incomplete. The line/fill plot in Fig.2 with the lakes' means and standard deviations refers to all the polygons shown in Fig. 1. Analogously, the snow and ice polygons plotted in Fig. 2 are the ones already highlighted in Fig. 1. This information is briefly contained in the caption of Fig. 2, but perhaps the amount of data (number of lakes and snow/ice samples) is not explicit. We will detail the caption of Fig. 2 further in the revised manuscript. It is instead true that the Amery series have mixed 6-day and 12-days coherence. More specifically, the first pair (04-Jan-2017 -> 16-Jan-2017) is separated by 12 days, whereas the rest of the Amery acquisitions is characterized by 6-days repeat. We will make sure to clarify this in Table 1. In conclusion, although we agree with the reviewer that the mixing 6 days and 12 days coherences should be done with caution, we believe that in the case of Amery, the impact of this temporal heterogeneity is negligible. Also, in interpreting the coherence differences between Amery and RBIS, we highlighted the fact that the latter have lower values also due to the longer (12 days) interval (lines 144-145).

The second part of the comment is not clear to us. Regarding the timespan of the plots in Fig. 2 for the two different areas, we believe that they are coherent with the dates specified in Table 1. The first and third plot refer the Amery, whereas the second and the forth refer to RBIS.

*4) Are there any different characteristics in amplitude/coherence/phase between supraglacial and englacial lake?*

Sentinel-1 has only limited penetration depth of several meters, so only shallow englacial lakes can be detected. However, our time series of Fig. 7 might actually be some sort of englacial lake (by developing a frozen lid). And we will discuss this aspect better in the discussion.

*Technical correction:*
*Line 113, 'the results', please be more specific.*

It refers to Fig. 2. We will clarify this in the revised version.

*Fig 2. Please provide complete legends for subplots 2-4. Fig caption are not complete. It only takes about the time series, but not the specific amery a, b, d and RBIS examples.*

The examples are labelled by the legend. We will try to be more complete and the caption will be extended.

---

## Author Response (AR1)

**Response to Referees on tc-2021-169**

We appreciate the reviews and comments from both Referees. Please find the response to Referee 1 on pages 1-17, and the response to Referee 2 on pages 18-20.

**Response to Referee 1 on tc-2021-169**

First, we would like to thank the Referee for reviewing and commenting the manuscript, which improves the quality of the manuscript. Please find the item-by-item reply below, with the original comments in *italics* and the responses in blue. The suggested changes have been implemented in the revised text.

***General Comments***
*My main concern pertains to the overall readability of the manuscript. While it is clear that the authors understand the background literature, methods, key results and implications of their research, their writing style is in general unconventional for a piece of scientific writing, insofar as it is highly verbose, often grammatically and/or typographically incorrect and, hence, difficult to follow/comprehend from a reader's point of view. This is further confounded by what appears to be inconsistencies in the clarity/style of writing adopted in different sections of the manuscript, missing information needed to fully understand the datasets and/or logic of arguments presented, and the occasional lack of relevant citations throughout the text (see my specific and technical comments for further details).*
*To address these issues, I recommend that all coauthors take the time to carefully restructure the wording of the manuscript to: a) more logically explain (and justify fully) the choice of all techniques and methodological decisions used/taken, b) correct typographical/grammatical errors and, c), cut down and hence improve the overall focus/narrative of the text. To assist the authors in this regard, I have made some suggestions on how the first two paragraphs of the introduction could be rewritten (see bottom of this document). If needed, the authors may also find the following resource (https://www.the-cryosphere.net/submission.html#english) and links therein helpful.*

We appreciate the suggestions and the text has been modified according to the specific comments. Generally, changes that are made include:
- Data and method section is elaborated.
- En dashes instead of hyphens are used to show time/value ranges.
- Verbose/incorrect/redundant sentences are rewritten/removed.
- Redundant definite articles have been moved.

***Title.*** *'InSAR' is an abbreviation and hence inappropriate for use in a title. Suggest rephrasing title to: 'The potential of synthetic aperture radar interferometry for assessing meltwater lake dynamics on Antarctic ice shelves' or similar instead. (see also my comments regarding Line 246).*

This has been changed in the revised text (in this case the title).

***Line 2.*** *Suggest replacing 'Yet' with 'Despite these phenomena'. Replace 'or' with 'and'.*

This has been implemented in the revised text. All comments regarding rephrasing/grammatical corrections will be replied as 'done' in the following text of this reply.

**L3.** *Suggest either ending sentence after 'limited', or briefly explaining what the limitations of optical satellite imagery are here.*

Added 'during polar night and in cloudy conditions' after 'optical satellite imagery'.

**L8.** *Change 'The analysis' to 'Our analysis'. At end of this sentence, change 'confounded' to either 'hard to distinguish' or 'indistinguishable'. Then change next sentence to: 'Despite this finding, we show using a combination of backscatter and InSAR observations that lake dynamics can be effectively captured during other non-summertime months'.*

(Now L9) Done.

**L11.** *Sentence beginning 'In particular'. For conciseness, suggest merging this and next sentence to: 'Moreover, our findings highlight the utility of InSAR-based observations for discriminating between refrozen ice and subsurface meltwater, and indicate the potential for phase-based detection and monitoring of rapid meltwater drainage events'.*

(Now L12) Done.

**L56.** *Remove 'The' and begin sentence 'Coherence is considered an indicator of changes'. Note also that phase difference does not correspond to the 'average' difference, but a very precise measurement of whole wavelength (and some fractional component) range difference. Suggest editing the rest of the sentence to reflect this.*

(Now L43) Done.

**L59.** *Sentence beginning 'This combination'. Who expects this? Add a reference to back up this claim, otherwise say 'We expect this' or similar.*

(Now L48) Changed to 'We expect this combination…'

**L67.** *Suggest changing to: "However, the value added using InSAR for such applications has not yet been examined".*

(Now L52) Done.

**L68-71.** *The structure of this paragraph is rather difficult to follow. Suggest beginning with "In this paper, we assess the potential of C-band backscatter and InSAR data to ... For this purpose, we use a combination of backscatter, coherence and phase information to monitor recent meltwater features over two East Antarctic locartions – the Amery and Roi Baudouin (RBIS) ice shelves – using data collected by Sentinel-1a/b in 2017/2018. To supplement the interpretation of our (In)SAR-based analyses, we also utilize spatially and temporally collocated optical satellite data.*

(Now L54-58) The whole paragraph has been changed as suggested.

*L75. Sentence is very long and could be split in two after reference to Lenaerts et al. (2016).*

(Now L63) Paragraph has been rewritten.

*L81. Why was a lake dataset not available? The reader shouldn't need to guess this, so state explicitly here. (I think you mean that no previously published dataset exists?). When mentioning Landsat data, also point readers to Section 2.2 for reference.*

(Now L68) Changed from 'a reference lake data set was not available' to 'no previously published dataset from in situ studies is available'. Reference to Section 2.2 is added.

*L83. Sentence beginning 'Our lake class'. If my understanding of the above is correct, then this sentence is confusing as it suggests that a preexisting lake dataset does indeed exist. If so, why didn't you use that here? Please either edit sentence of clarify or remove from the text.*

Previous studies observed the Amery Ice Shelf with satellite imagery, but no in situ dataset is available. A modification in Line 68 has been implemented to clarify this.

*Figure 1. This is a nice figure, but please add latitude/longitude information to each panel so that readers can easily deduce locations. Please also consider showing two additional, zoomed out panels showing the location of each subset within both ice shelves, and (if necessary) in these and all pre-existing panels, add the ice sheet grounding line for reference (e.g. https://doi.org/10.7280/D1VD6G). (I know that the lower right panel shows this to a certain extent, but it's difficult to see details. In general, I also find the labels rather small and difficult to locate, so these could also be enlarged (plenty of space on figures to do this).*

The suggestions have been implemented. After adding the extra panels it is indeed clearer, but the pre-existing panels become too small. We hope it is still readable after the modification.

*Figure 1 caption. Please define RBIS in full in caption. Replace 'close ups' with 'inset' or 'detail'. Insert comma after 'panels'. Change 'delineated in black curves' to 'delineated as black curves'. In next sentence, what does 'indices' refer to? Labels? If so use 'labels' instead for clarity. Suggest rephrasing following sentence to read: "...are also delineated for comparison against backscatter intensity and coherence values observed over lakes (Fig. 2)' or similar. For general readability, next sentences could/should read: 'Panel R2 illustrates the lake feature shown in Figure [insert number here]. Inset shows location of the analyzed locations'. Please also state in caption which band/band combinations are shown (Landsat).*

Done.

*Table 1 caption. Should read '... used in this study'.*

Done.

**L99.** *Please add more information (and references if necessary) on how the images were denoised, calibrated and corrected here. (The reader shouldn't have to look it up for themselves).*

(Now L93-98) Information on calibrated top-of-atmosphere (TOA) reflectance (Chander, 2009) Landsat images has been added in the revised script.

**L92.** *I presume this is a typo and should say 10x10 m resolution (i.e. the native resolution of IW GRDH; https://sentinels.copernicus.eu/web/sentinel/technical-guides/sentinel-1-sar/products-algorithms/level-1-algorithms/ground-range-detected/iw)? If not, then how does this impact the rest of your backscatter analyses?*

*Also, if my understanding is correct then NRCS is identical to radiometrically calibrated, sigma-nought (σ0) backscatter imagery. Sigma-nought backscatter is the much more commonly adopted term in the geosciences (RCS more so in engineering), so given the likely readership of The Cryosphere, I would instead refer to σ0 in place of NCRS here and universally throughout the manuscript. Following this point, be careful that the imagery you downloaded from Google Earth Engine isn't already in sigma-nought format, as the 'radiometric calibration' you mention GEE perform above may imply.*

About the resolution, the Single Look Complex images of Sentinel-1 have a 20 m (azimuth) x 4.5 m (ground range). The GRDH are derived by averaging around 4.4 looks in range to make the resolution approximately 20 x 20 m. The products are then provided in an upsampled format, with pixel spacing/posting of 10 m. So we believe that the value of 20 m reported in the manuscript (that is the relevant one for the equivalent number of looks, or else for the radiometric precision when averaging areas of distributed targets such as ice, snow and water) is correct.

The resolution does not really affect the analyses. We specify the resolutions to clearly introduce the datasets. This could be removed if it is not necessary.

We appreciate the advice of using a consistent nomenclature throughout the manuscript. In the revised version 'NRCS' is therefore changed into 'σ0', including the figures. About the calibration, our understanding (see https://developers.google.com/earth-engine/datasets/catalog/COPERNICUS_S1_GRD) is that GEE already provides radiometrically calibrated GRD images, and, as such, in σ0 format.

**L95.** *Confusing. If you have multi-looked the SLC to create your own GRD/Sigma-nought imagery then why did you bother downloading GRD imagery from GEE? Clarify here.*

The main difference between our products and the imagery downloaded from GEE is that our analysis on the RBIS focuses on ascending track 59, whose data were not available on the rolling archive of Sentinel Scihub before July 25, 2017. To analyse the lake formation since 2016 (shown in Figure 3), we conveniently made use of the imagery available on GEE. This has been clarified in L75-77 of the revised manuscript.

**L97.** *Why is a geoid only used for Amery? I expect this is a typo?*

At the moment of processing we lacked a detailed DEM for Amery and therefore used the Geoid as this closely resembles the ice shelf surface. Although we agree that this could be improved (e.g. by using REMA), we expect it will not change any of the conclusions of the analyses. This is proven in Fig. 4 and Fig. 7, as the meltwater features in the InSAR images are in the same location as in Landsat images. The use of DEMs has been detailed in L89-92.

*L98+ This is a good example of missing methods I mentioned in my general comments. Here, I am surprised to see absolutely no information about how the authors generated their interferograms. This must be added, including, as a minimum, information on e.g. temporal/perpendicular baselines used, type of processing performed (I assume single-pass DInSAR?) and (if used) any DEM used to remove topographical phase.*

The overview of DORIS has been added in the revised manuscript (L83+ and new Fig. 2).

*L99-100. Again, for clarity/reproducibility purposes, much more info should be included here on e.g. the band/band combinations used, pixel resolutions of the data, and any relevant pre-processing steps you applied to your Landsat imagery. Stating that you simply downloaded them off GEE is not appropriate for a scientific paper.*

TOA Tier 1 Landsat surface reflectance data (Chander) of bands (RGB) were downloaded from GEE at their native 30 m pixel resolution without any additional pre-processing steps. This has been added in the revised manuscript (now L93-98).

*L105. I think it's important to explicitly state here that for each class type analyzed, you calculated the mean and standard deviation backscatter for all observed features. As written this is not obvious, and leads to confusion over what the difference is between 'lake class' and the 'individual lakes' (for this reader at least ...). This could also be made more explicit in the caption of Figure 2.*

(Now L109) The sentence has been completed as 'For this purpose, the temporal variations in $\sigma0$ and coherence are compared per lake, snow, ice class by analysing their time series of the mean and standard deviation for each class (i.e. lakes, snow and ice).' Also added 'Mean and standard deviation are calculated for all features presented.' to the caption.

*L106. For clarity, 'mono-dimensional' should be changed to 'cross-sectional'.*

(Now L112) Done.

*L107. Repetition of coherence, NCRS and phase. Suggest restructuring sentence to avoid this.*

(Now L114) Changed from '…combining NRCS, coherence and phase information...' to '...combining SAR backscatter intensity with InSAR information...'

*L112. Suggest changing to: 'The mean sigma-nought timeseries of lakes, snow and ice (cf. Section 2.2) display strong seasonal variability, consistent with the changing nature of both*

*surface snow and ice properties and the evolution of supraglacial lakes through time (Figure 2). On Amery Ice Shelf, our observations reveal...'.*

(Now L127) Done.

*L113. This is written in an odd manner which implies that snow transforms into lakes and then ice, which is not what the authors intend to say. What I think they mean to say is that snow, lakes and ice for the most part display different (though reasonably constant) backscatter properties throughout the year, with the exception of JF when the backscatter associated with snow and lakes fall rapidly.*

(Now L128) Now the sentence has been changed from 'The results on Amery ice shelf show that the NRCS decreases from snow (˜0 dB) to lakes (˜-5 dB) and ice (˜-10 dB) during fall, winter, spring…' to 'On Amery Ice Shelf, our observations reveal that the $\sigma 0$ has different levels for snow ~0 dB), lakes (~-5 dB) and ice (~-10 dB) and is relatively constant (fluctuations within ~1 dB)…'.

*L116-125. To shorten the text here, I question whether the authors even need to discuss (and, in Figure 2, show) the individual lakes because for the most part, the average of multiple mapped lakes makes seems to support their arguments just as well. In this regard the individual lake observations are a slight distraction from the overall story revealed by the class averages, so I think they could probably be removed. (as the authors show, there is significant variability from one lake to the next, so focusing in on specific lakes only serves to deviate from what's happening on the whole). This is also largely true for the coherence discussion of RBIS a and f in Section 3.2, as your later coherence images (Figure 7) in any case demonstrate the process of refreezing in a much more convincing way).*

(Now L132 onwards) We agree. Now the discussion is removed and the figure (now Fig. 3) has been modified by removing the curves of the individual lakes.

*L126-134. For clarity, suggest editing sentence to say '...of select cross-sectional transects. In the case of both RBIS 'a' and Amery 'd' (location shown in Figure 1), for example, backscatter timeseries show significant inter-annual variation (Figure 3)'.*

(Now L135-137) Done.

*L129. 'Border of low NRCS and inner areas of high NRCS'. Revise this sentence to explicitly state that this refers to the edge and central regions of the lake, respectively.*

(Now L139) Changed from 'After this, a clear spatial pattern emerges with borders of low NRCS and inner areas of high NRCS, followed again by a new area wide decrease in NRCS in the Dec. 2017 - Jan. 2018 melting season.' to 'Subsequently, a clear spatial pattern emerges with borders of low $\sigma 0$ at the edges and high $\sigma 0$ in the central regions, which respectively refer to the edge and central regions of the lake. This pattern is followed again by a new area-wide decrease in $\sigma 0$ in the Dec. 2017 - Jan. 2018 melting season.'

***L136-152.*** *These are clear, well written paragraphs. They are logical and concise, and could be considered a model for how the rest of the manuscript should be written.*

(Now L146-160) We appreciate the suggestion.

***Figure 2.*** *See my comments regarding L116-125 above. If the authors choose to retain the analysis of individual lake features, then they should make the lines thicker as these are currently very difficult to see both on-screen and in print. To enhance visibility of these lines, I'd also consider making the standard deviation ribbons more transparent as these currently dominate/clutter the figure. As per Figure 1, I also think the labels (and legend especially) should be made bigger/more prominent.*

*For the coherence plots (and to a lesser degree sigma-nought), I wonder if the high frequency variability discussed by the authors could be smoothed out using something like a running mean? While this variability is interesting, it's a little distracting, and is later largely ignored in the text anyway.*

(Now Figure 3) The individual curves are removed from the figure. However, we decided not to smooth the curve, because the gaps in the data become the obstacle of computing the running mean. If there were no such high frequency variability, filling the gaps via interpolation would be a nice idea. But due to this variability, interpolating will also be unreliable.

***Figure 2 caption.*** *Sentence should read '... over the Amery and Roi Baudouin ice shelves (see Fig. 1 for locations). Change 'Moments ...' with 'Times with a lack of 6/12-day...'.*

(Now Figure 3) Done.

***Figure 3.*** *As per Figs 1 and 3, please make all labels larger. Please also add lat/longs to both maps along with scalebars. Also consider zooming both images to show more detail over lakes (bottom right panel especially).*

(Now Figure 4) This has been implemented to the revised script.

***Figure 4.*** *Nice figure! As above regarding label size.*

(Now Figure 7) Thanks for the suggestion. Label size has been increased and location and scale bar have been added.

***Figure 4 caption.*** *Remove 'synoptic' (incorrect usage in this context) in first sentence, and cross-reference Fig. 1 for location at end. In next sentence, remove 'the' proceeding coherence, and add '... and resulting phase difference interferograms are shown ...'. In next sentence, should say 'The high frequency fringes surrounding each lake represent a convolution of both ice flow and tidal motion'. In the last sentence, please state which band/band combinations are shown (Landsat).*

(Now Figure 7) Done. Thanks for the correction.

***Figure 5.*** *Really nice figure, but please add lat/longs and scalebars.*

(Now Figure 8) These have been added.

***Figure 5 caption.*** *Unnecessary use of 'right' which should be removed. In the next sentence, change 'hereby' to 'hereafter'. In the following sentence, change 'reported' to 'shown (right panels)'. Please also state which band/band combinations are shown.*

(Now Figure 8) Done.

***L147.*** *I think this should say 'between Oct. 2017 and Jan. 2018'. Change 'polygons' to 'surveyed snow, ice and lake areas'.*

(Now L158) This has been corrected.

***L154.*** *Amery Ice Shelf.*

(Now L163) Done.

***L159.*** *Replace 'brighter' with 'greater'.*

(Now L168) Done.

***L172.*** *Why 'possibly'? Provide evidence to justify claims here. (Also, RBIS is a rather slow flowing ice shelf, so horizontal displacement should not influence phase coherence over 12 days as much as one might think (see Mohajerani et al. (2021) who were able to map GLs across this region Antarctica with good coverage using double difference InSAR. This technique requires almost perfect coherence, suggesting 12 days is more than sufficient here).*

*In the next sentence (beginning 'In Oct. 17...'), I think better referencing to Figures 2 and 5 is needed as I don't see any change in coherence from Figure 5 alone.*

First, we would like to clarify that here it is a mistake that 'coherence in this region drops in winter', as it should be summer in Antarctica. To justify the observations, we used wind and precipitation from ERA5, and derived melt factor with radiometer data (SSMIS). The data and methods have now been described in L99-105, and L117-124. The figures (new Fig. 5 and Fig. 6) together show that the coherence drop can be due to precipitation in Oct.–Nov., and due to melt in Dec.–Jan. Therefore, it is true that claiming coherence drop is due to the 12-day revisit time is not discreet, and we have modified the text (now L184-186) to better analyze this.

Reference to the figures has been added.

***L176.*** *Change tense to be consistent with the rest of the paragraph. Also, while what you go on to say in Lines 176-177 is technically true, visually I can't tell the difference between the lakes you are discussing and the drainage network. Suggest rewriting this sentence for clarity*

*to specifically emphasize the observed change from a lake to a (presumably) connected drainage network through time.*

*This is a really nice observation by the way, demonstrating in a compelling manner the utility of coherence to see what simple optical and/or backscatter images cannot.*

(Now L188) The tense has been corrected. Changed from '…which is not straightforward to see in the NRCS or optical imagery. This highlights the increased potential for coherence over the backscatter intensity in delineating the lake network.' to '...which is shown as dark strips between the highlighted lakes in the lower middle panel of Fig. 8. The patterns are clearly newly formed compared to the lower left panel of Fig. 8. This change is not straightforward to see in the $\sigma0$ or optical imagery.'

**L179.** *Suggest rewriting to begin: 'Interferometric phase difference maps (Figure 4) emphasize... Amery ice Shelf.*

(Now L194) Done.

**L180.** *Initially I didn't see any fringes you refer to (c/w Amery c for example) given the dominance of the high frequency (ice flow) fringes surrounding the lakes, but then I realized you meant the very low frequency fringes on the lakes themselves (~1 cycle of -π to π only). I suggest you state this more clearly (and perhaps label the figure accordingly) so that readers don't incorrectly focus in on the high frequency fringes.*

(Now L195) The low-frequency fringes in the lake centre and the high-frequency fringes surrounding the lakes have been specified in the revised text.

**L182.** *Edges of what? I can work out what you mean, but this can be written more clearly for ease of reading. Possibly also consider citing appropriate figures and panels.*

(Now L197) Edges of lake Amery b. This has been added to the sentence. And the whole sentence has been changed to 'This pattern of discontinuity is consistent with lower coherence at the edges of lake Amery b, which most clearly follows the orange delineation curve in the Oct. 2017 coherence panel of Fig. 7.'

**L183.** *Suggest writing as '...... increase through time indicates the presence of lakes until October 2017, followed... in November of that year. Consistent with our InSAR-based observations Landsat ...'.*

(Now L199) Done.

**L179-203.** *In general, this is another clearly written and easy to comprehend series of paragraphs compared with the earlier section of the manuscript.*

(Now L194-224) Thank you for the suggestion.

*L179-186. Regarding Amery Ice Shelf, what (if anything) can we learn about the detection of the hydrological network that is clearly visible in Figure 4 (top row), and which disappears after March 2017 in the coherence images? (suggesting formation between Mar 11 and 17th and persistent presence (freezing?) thereafter). This is a visually striking feature in the center of these panels that I was surprised to see no discussion of here and/or in Section 3.2.*

We appreciate the suggestion, and the discussion has been added in L177-179 of the revised manuscript as 'Moreover, between Amery b and c, a hydrological network that is clearly visible as high $\sigma0$ in the $\sigma0$ panels is present only in the Mar. 2017 coherence panel as low coherence. This could suggest the surface refreezing between Mar. and Jul. 2017, similar to that discussed by Antonova et al. (2016).'

*L187. I'm not quite sure I follow this, as the color scheme always goes from blue to blue. Suggested rewriting for clarity.*

(Now L205) This has been spcified as 'from right to left of the Dec. phase image, fringes change from red--blue--green--yellow to red--yellow--green--blue, forming a whirl-like feature'.

*L198. How big was this uplift? I think that would be a valuable addition here, and can be estimated either through unwrapping the phase or counting the fringes.*

We agree that it would indeed be very interesting. By counting the fringes, the result is approximately 7 fringes, each measuring 2.8 cm in the line-of-sight. Assuming a vertical movement, this corresponds to an uplift of approximately 24 cm (taking into account an incidence angle of approximately 35°). However, without data for validation, we were cautious in providing an exact number.

*L190. And presumably some tidal component, as \*I think\* tides haven't been removed? (see also my comments on the omission of any methods detailing exact InSAR processing above).*

(L209) We agree. This has been added to the revised script.

*L194-203. Great series of observations.*

(Now L213-224) Thank you.

*Figure 6. Nice figure, but please add lat/longs and scalebars to all panels.*

(Now Figure 9) Since all images are in the same scale, we added lat/longs and scalebar to the Landsat panel for simplicity.

*Figure 6 caption. Replace 'interferometric phases' with 'interferometric phase'. For brevity, suggest rewording next sentence as 'Two near-contemporaneous Landsat 8 panchromatic (band 8) images are also shown (right panels)'.*

(Now Figure 9) Done.

**L201.** *Replace 'starting at the edges' with 'towards the center of the lake' or similar.*

(Now L221) Done.

**Figure 7.** *Very nice series of observations! Enlarge labels and add lat/longs and scalebar, though.*

(Now Figure 10) This has been implemented.

**Figure 7 caption.** *Replace 'interferogram phases' with 'interferograms' or similar. Please also state which band/band combinations are shown for Landsat imagery.*

(Now Figure 10) Done.

**L204.** *I'm not sure I completely follow what you're trying to say here, as the sentence contains a grammatical error. Suggest rephrasing for greater clarity.*

(Now L222) Changed from 'The interferogram shows similar results here, but with the added value that the interpretation of high-low backscatter compared to the surroundings is less ambiguous.' to 'The interferogram shows similar results here. However, compared to interpreting the refreezing of the lake solely based on backscatter intensity, adding interferograms to the observation helps reduce ambiguities in the interpretation.'

**L205.** *Suggest beginning this section like: 'Using SAR-based observations acquired across two East Antarctic ice shelves, we present evidence of the utility of backscatter ...'.*

(Now L226) Changed into 'Using SAR-based observations acquired across two East Antarctic ice shelves, this study presents evidence of the utility of backscatter intensity and coherence to assess meltwater lake dynamics.'

**L213.** *Change 'Coherence' to 'Interferometric coherence'.*

(Now L234) Done.

**L215.** *And all other types of SAR SLC data ... not just that acquired by Sentinel-1.*

(Now L236) Sentinel-1 has been removed from the sentence.

**L222.** *I think its important to stress here that low coherence isn't just about refreezing (or not). Radar waves are fully attenuated by water, so you will always get poor coherence as long as there is water. The authors should rephrase this sentence to reflect this point.*

(Now L243) This has been changed from '…while meltwater lakes show a low coherence due to the constantly changing ice/water interface', to '…while meltwater lakes show a low coherence due to the constantly changing ice–water interface and the increased attenuation due to the presence of water'.

*L229. This sentence may lead to confusion as it implies water volumes can be calculated using InSAR techniques. Suggest rephrasing to articulate the intended point more clearly.*

(Now 249) We intended to say that estimating the water volumes is not within the scope of this study. To avoid confusion, this sentence has been removed.

*L232. For consistency with the text above, suggest changing to 'affected by tidal and horizontal motion'.*

(Now L252) Done.

*L237. Again, I think it'd be really nice to see an estimate of the uplift here, derived from either fringe counting or unwrapping the phase (see also my comments on L198).*

The reply is the same as for L198.

*L241. Amery Ice Shelf.*

(Now L261) Done.

*L243. Argument regarding line-of-sight observations only. This is actually only true for Sentinel-1 which, at present, only has one look direction over these ice shelves. Sentinel-1 (or any other sensor for that matter) collected in both ascending and descending orbit could deconvolve those parameters potentially yielding a better impression of subsidence/uplift, or at the very least a different (and possibly validatory) view of the lake dynamics relative to that gleaned from a single look direction.*

*Suggest rephrasing the sentence to stress these points, and refocus the sentence away from Sentinel-1 'only' towards a more broad discussion of the different SAR sensors that could possibly be used.*

(Now L268) Added 'With SAR acquisitions from sensors in both ascending and descending orbits, it is however possible to better quantify the lake subsidence/uplift.' to the paragraph.

*L244. I wonder to what extent this sentence is true, since more complicated processing techniques like double-difference InSAR (e.g. Mohajerani et al.; 2021) could presumably help to cancel out ice flow signals. De-tiding observations using a tidal model could also remove vertical motion due to tide (see, for example, MacMillan et al., 2012). Did the authors investigate the applicability of these techniques for improving signal-to-noise over the lake areas? (I'm not suggesting this necessarily needs to be done if not, but I feel a more nuanced/careful discussion of how ice/tide displacement could possibly be mitigated to lake detection easier should be included here).*

(Now L265) This discussion has been included as 'Second, the interpretation of phase change should be done relative to the displacement of the lake surroundings in the line-of-sight. As the meltwater lakes typically develop on locations with strong ice and/or tidal displacement, interpretation should be done relative to that displacement. Therefore, to better derive the

exact height change of lake ice lids, additional processing is needed to cancel out ice movements (Mohajerani et al., 2021) and to filter out signals due to tidal movements (McMillan et al., 2012).'

*L246. To conclude this section, I think there's big scope to include one or two sentences on the potential advantages of 'next-generation SAR' remote sensing capabilities for lake monitoring. This could involve a discussion of the <6-day imaging capabilities afforded by the launch of Sentinel-1c (~2022), and/or the upcoming (2023) launch of the NASA-ISRO SAR mission (NISAR). While the latter will have a repeat pass time of 12 days over the polar regions, its dual-wavelength (L- and S-band) imaging capabilities may have good potential to circumvent confounding issues such as snow blow and other atmospheric effects, quantify thin/forming ice lid thicknesses etc.*

*If the authors do not wish add such a discussion, then I recommend editing the title of the study to be sensor specific, e.g. 'The potential of Sentinel-1a/b synthetic aperture radar interferometry for assessing meltwater lake dynamics on Antarctic ice shelves'. (see also my comments regarding the title, L243 and L261-265 above).*

This has been added to the discussion (Now L270-274).

*L248. I think this sentence could (and should) be snapper. Suggest rephrasing to 'This study has provided insight into the utility of InSAR for monitoring meltwater lake dynamics' or similar.*

(Now L276) Done.

*L261-265. This is largely repetition of Lines 238-246 which I think can probably be significantly shortened and merged with Lines 266-268. Suggest something like: 'Despite noted limitations to current Sentinel-1 InSAR imaging over parts of Antarctica, we show that InSAR provides promising potential for monitoring meltwater lake dynamics beyond that afforded by conventional, backscatter-only, analyses. Such potential could pave the way for …'.*

(Now L290-292) Done.

*Referencing. I have noticed multiple inconsistencies in the manuscript. Please ensure referencing style is consistent throughout and adheres to The Cryosphere's specific referencing format (https://www.the-cryosphere.net/submission.html#manuscriptcomposition).*

Thanks for the suggestion. This will be double-checked in the revised script.

*L5. Incorrect grammar and sentence structure. Suggest rephrasing to: 'In two case study regions over the Amery and Roi Baudouin ice shelves, East Antarctica, we examine spatial and temporal variations in SAR backscatter intensity and interferometric (InSAR) coherence and phase over several lakes derived from Sentinel-1a/b C-band SAR imagery.*

(Now L6) Done.

**L15.** *Insert commas before and after 'however'.*

(Now L14) Done.

**L55.** *'By a certain time' is colloquial. Suggest 'by a particular temporal baseline' or similar instead.*

(Now L42) Done.

**L65.** *Remove 'basically' (colloquial usage inappropriate for scientific writing).*

(Now L51) Done.

**L75-79.** *These sentences are repetitive and could easily be merged for conciseness. Also, in the last sentence, I think it's important to explicitly state why you delineated polygons of surrounding snow and ice, as this is unclear.*

(Now L61-66) Added '…manually delineated sample polygons of snow and ice surfaces based on Landsat imagery for studying the difference between meltwater lakes and the solid surrounding regions'.

**L88.** *Insert comma after 'For both products'. At end of sentence, also add citation to back up this statement.*

(Now L77) Done. This statement (that only HH-polarisation is available) comes from our random searching over several locations in Antarctica. For reliability, we have changed 'Antarctica' into 'the studied ice shelves'.

**L99.** *Pronouns are not to be preceded by 'the', so remove 'the' before Google Earth Engine. (Also true for the likes of 'the Amery Ice Shelf', 'coherence' etc.).*

(Now L96) Thanks for the correction.

**L102.** *Add comma after 'dynamics'. At end of sentence, explicitly state where you perform this analysis (i.e. over the lakes and control (snow/ice) sites). For clarity, this should probably also involve merging the following sentence.*

(Now L107) Done.

**L105.** *Insert comma after 'purpose'. (Note: punctuation errors of this type are a recurring issue and one that I encourage the authors to carefully correct for throughout the manuscript).*

(Now L109) Done.

**L136.** *'Amery ice shelf' is a pronoun and so should be capitalized. Note that this correction should be carefully applied to all pronouns in the manuscript.*

(Now L128) This has been corrected.

**L150.** *Insert commas before and after 'however'.*

The discussion of the time series of individual lakes has been removed.

**L166.** *Insert comma after 'gradual'. Regarding the next sentence, I suggest also labelling the circular feature you refer to in the figure, as it took me a while to recognize exactly what you mean.*

(Now L174) Done. An arrow to the feature has been added.

**L169.** *Reference Fig. 5 in the first sentence. The second sentence is also grammatically incorrect and should be edited to state that routine Sentinel-1 coverage commenced in 2017 and to date only acquires data with a repeat-pass of 12 days.*

(Now L180) Reference has been added. Changed sentence from 'Since the Sentinel-1 SLC temporal coverage is lower than for Amery, SLC coverage only started in July 2017 (Fig. 5).' to 'Differently from data on Amery Ice Shelf, the Sentinel-1 SLC acquisition only started in July 2017, with a 12-day revisit (Fig. 5).'

**L172.** *Should read '..., with only intermediate sigma-nought values'.*

(Now L183) Done.

**L214.** *Should say 'assess'.*

(Now L235) Done.

**L215.** *Remove 'such as Sentinel-1'.*

(Now L236) Done.

**L228.** *To maintain the flow of the text here, suggest rephrasing this sentence to: 'Beyond coherence, we also demonstrate the potential of interferometric phase for assessing ... in areas of high coherence'.*

(Now L249) Done.

**L231.** *Suggest changing 'instant' to 'rapid (sub-weekly) meltwater events', since changes over 6 days can hardly be classified as instant.*

(Now L250) Done.

**L233.** *I think this should say '...an easier detection of stable ice and lake refreezing than coherence and backscatter intensity ...'?*

(Now L252) This has been corrected.

*L235. Incorrect grammar/sentence tense. Suggest rewording to: "While InSAR-based techniques show clear potential for monitoring meltwater lake evolution, there are several key limitations associated with this technique compared with conventional optical- and SAR backscatter-based imaging. First, InSAR requires …'.*

(Now L257) Done.

*L240. Replace 'may' with 'can'.*

(Now L261) Done.

*L241. 'day' should read 'days'. Also, suggest rewording 'Due to this difference' to 'Due to these differing imaging times' or similar.*

(Now L262) Done.

*L253. Sentence beginning 'A generalization'. Reword to 'We show that meltwater detection using backscatter is, however, not straightforward, as meltwater lakes often …'.*

(Now L281) Done.

*L255. Replace 'context' with 'circumstance'. Also suggest removing 'i.e. the coherence and interferogram phases'. (this is unneeded technical info for the conclusion).*

*Above, the authors could also consider rephrasing the text to offer a more well-rounded discussion on the application of SAR in general, rather than specific application of Sentinel-1 data (see my comments regarding title, L243 and L246).*

(Now L283) Done.

*L256. 'Besides' should not be used to begin a sentence. Replace with: 'In addition, we show that InSAR-derived information can also be used to observe meltwater lake evolution (and potential drainage) with high accuracy beyond that afforded by conventional backscatter or optical satellite imaging' or similar. Then begin next sentence with: 'Specifically, InSAR coherence information allows for the detection of changes in the …, while interferometric phase can effectively track the spatial and temporal evolution of ice refreezing. Maps of interferometric phase moreover allow for the detection of abrupt lake drainage (or filling) events via changes in the relative displacement of the surface between successive SAR passes'.*

(Now L284) Done.

*L274. I think this should say 'WL was responsible … processing and analyzing the results …'.*

(Now L298) Done.

***L278.*** *Remove NSF-OPP awards and rest of lines 279 and 280 as these are not relevant to this study.*

That was the standard citation format required by the publisher (please refer to the PGC acknowledgement site https://www.pgc.umn.edu/guides/user-services/acknowledgement-policy/).

***L322.*** *Please cite final (non-TCD) publication.*

This paper gives the following information: Review status: this preprint was under review for the journal TC. A revision for further review has not been submitted. Therefore, this citation has been removed.

**References**

Chander, G., Markham, B. L., and Helder, D. L.: Summary of current radiometric calibration coefficients for Landsat MSS, TM, ETM+, and EO-1 ALI sensors, Remote Sensing of Environment, 113, 893–903, https://doi.org/https://doi.org/10.1016/j.rse.2009.01.007, 2009.

**Response to Referee 2 on tc-2021-169**

Thank you for reviewing and commenting the manuscript. Please find the item-by-item reply below, with the original comments in *italics* and the responses in blue.

*General comments: This paper evaluated the beneficial of combining SAR amplitude, InSAR coherence and phase information for meltwater lake dynamics. The topic fits well with The Cryosphere journal and it provides useful information for investigation for lake dynamics in Antarctic environment. The selected cases over Amery and Roi Bauouin ice shelves (RBIS) shows that SAR amplitude, InSAR coherence and phase are complimentary for lake dynamics monitoring. However, I think the presented examples may oversimplify the interpretation of SAR amplitude, coherence and phase for monitoring meltwater lake dynamics, as we know other factors, other than seasonal melting-refreeze process, such as weather event (snow, rainfall, et,al) and sensor acquisition geometry (descending/ascending) could also affect amplitude/coherence/phase variation. There are also some other issues with this paper, such as convincing evidence about the lake status in the analysis, and incomplete/confusing data information that were used in the study.*

We would like to thank Referee 2 for the positive comments on the potential. We agree with the arguments that other factors such as changes in snow properties and sensor effects also may impact the signals. This was in fact mentioned already in the manusript, but it was perhaps not given the proper weight. In the revised version, we have added precipitation data and melt estimations in the assessment, and possible improvements based on sensor acquisition geometry in the discussion.

*Specific comments*
*1) Instead of just few selected data, please provide a complete time series amplitude, coherence and phase analysis for the cases in Fig 4 & Fig 5. I think this would still show the benefits of different information (amplitude, coherence and phase), but it would provide a more objective sense/perspective for reader to understand potential drawbacks of each different information. Incomplete data also make some of the statements confusing in the paper. For example, Line 155-157, it talked about amplitude/coherence for summer melting, but there are no SAR data shown in the Fig 4.*

Starting from lines 155-157, we perhaps caused confusion by mentioning the melt of the ice. The lines refer to the background blue ice area, rather than the lakes. And the blue ice features are shown in the second panel at the bottom of Fig. 4 (RGB bands of Landsat image). However, it is true that our discussion of melt could be improved by providing a climate/melt time series (added in methods, Fig. 5 and Fig. 6).

As for the complete time series, it has 60 days of acquisition, and showing all the backscatter intensity, coherence and phase images will result in 3*60=180 images. This is not ideal to show in one figure. However, the selected images can already show characteristics under different distinct circumstances.

*2) Please provide evidence when refer to melting/refreeze/frozen status of the lake to make your statement convincing. For example, the authors explained the decorrelation in Jan 2017*

*data is due to melting (Line 195-196) in Fig 7. However, in this same figure, we see the Jan 2018 shows very good coherence and phase pattern. I would assume the area would be in similar freeze/melt status at approximate same time of different years. I am not sure whether the low coherence in Jan 2017 is due to melting or maybe other weather events. I think it would be helpful to collect some other information, such as temperature information from other sources, to support your statement. For all other data analysis, if it's possible to collect some external information such as temperature or optical imagery, I would suggest doing so that it's more convincing when you state its under melt or refreeze or frozen status.*

We appreciate the referee's suggestion in using other external data sources. It is difficult, however, to acquire completely concurrent Landsat images due to the cloud cover. We have therefore included ERA-5 data and radiometer-derived melt time series in the revised manuscript to show it more clearly. It is true that low coherence can be due to both melt and precipitation. This has been added in results and discussion.

*3) Incomplete data information. For the time series of mean and standard deviation over selected polygons mentioned in Fig 2, How many SAR data are used for this calculation and what are their acquisition times? are the mean and std for all the polygons shown in Fig 1? It might be helpful to provide complete data list in text or supplement. Are the coherence data for Amery all 12-days product? It would be not meaning to mix 6-days or 12 days data together to analyze lake-related information, as temporal difference would change that a lot.*
*Please show the outline of the sentinel-1 data in the last panel of Fig 1. What is the data coverage used in this study? Table 1 shows RBIS SLC data is from 2017/7/25—2018/4/15, however, in fig 2, the data coverage for RBIS is from 2017/1-2018/1, it is so confusing. Please provide accurate info for data you used. Also, the GRD and SLC time coverage are different as shown in fig 2, not sure how does this happen. I would assume you need to analyze amplitude/coherence/phase comparison for all data.*

We are sorry to hear that this information seems incomplete. The line/fill plot in Fig.2 with the lakes' means and standard deviations refers to all the polygons shown in Fig. 1. Analogously, the snow and ice polygons plotted in Fig. 3 are the ones already highlighted in Fig. 1. This information is briefly contained in the caption of Fig. 3, but perhaps the amount of data (number of lakes and snow/ice samples) is not explicit. We have detailed the caption of Fig. 3 and Table 1 further in the revised manuscript regarding the missing data. It is instead true that the Amery series have mixed 6-day and 12-days coherence. More specifically, the first pair (04-Jan-2017 -> 16-Jan-2017) is separated by 12 days, whereas the rest of the Amery acquisitions is characterized by 6-days repeat. We have clarified this in Table 1. In conclusion, although we agree with the reviewer that the mixing 6 days and 12 days coherences should be done with caution, we believe that in the case of Amery, the impact of this temporal heterogeneity is negligible. Also, in interpreting the coherence differences between Amery and RBIS, we highlighted the fact that the latter have lower values also due to the longer (12 days) interval (lines 151-156).

The second part of the comment is not clear to us. Regarding the time span of the plots in Fig. 3 for the two different areas, we believe that they are coherent with the dates specified in Table 1. The first and third plot refer the Amery, whereas the second and the forth refer to

RBIS. There is inconsistency regarding the GRD acquisition dates in Table 1 and Fig. 4, and this has been corrected.

*4) Are there any different characteristics in amplitude/coherence/phase between supraglacial and englacial lake?*

Sentinel-1 has only limited penetration depth of several meters, so only shallow englacial lakes can be detected. However, our time series of Fig. 10 might actually be some sort of englacial lake (by developing a frozen lid). In general, we expect longer wavelength SAR to provide improvements in this aspect, and this has been added to the discussion (L270+).

*Technical correction:*
*Line 113, 'the results', please be more specific.*

It refers to Fig. 2.

*Fig 2. Please provide complete legends for subplots 2-4. Fig caption are not complete. It only takes about the time series, but not the specific amery a, b, d and RBIS examples.*

(Now Fig. 3) The examples are labelled by the legend. The caption has been extended.

---

## Referee Report (RR1)

**Re-review of "*The potential of InSAR for assessing meltwater lake dynamics on Antarctic ice shelves*" by Weiran Li et al.**

I commend the authors for doing a fine job at addressing my concerns and for revising the manuscript accordingly. The manuscript is now much improved, and I believe that the addition of both the ERA5 precipitation and SSMIS passive microwave-derived melt estimates (added in response to some of Reviewer 2's concerns) are good supplements to the analyses presented. Below, I list several science and technical (mainly typos and grammatical) suggestions which should be incorporated into the final version of the manuscript. Line numbers refer to those in the revised manuscript. Once implemented, I believe the paper will be suitable for *The Cryosphere* and look forward to hopefully seeing it published in due course.

**Scientific Comments**

L45. Remove '*the whole or*' (phase will never be a complete integer component of the wavelength'.

L53. '*Radiometric*' isn't the correct word here since radiometry pertains to nearly all forms of remote sensing. Suggest '*optical and passive microwave satellite data*' instead.

L71. I appreciate the authors response regarding this paragraph, although I'm afraid the final sentence still doesn't make much sense to me. If the Spergel et al. dataset exists, why didn't you use it? Perhaps it was not publicly available at the time of analysis? If so, this should be mentioned here, otherwise the sentence should be revised to improve clarity or removed.

L76. Following the author's responses to this sentence, I suggest stating explicitly here the fact that no data exist over Track 59 of Amery because this isn't available on the Copernicus Open Access Hub. (This isn't explicitly obvious from Table 1).

L79. Sentence beginning '*The final backscatter products*'. Despite the author's responses to this sentence, I am still a little confused by this since the multi-looked cell size of the IW images on GEE are not 20x20 m (see https://developers.google.com/earth-engine/datasets/catalog/COPERNICUS_S1_GRD; "*This collection contains all of the GRD scenes. Each scene has one of 3 resolutions (10, 25 or 40 meters*").

I understand in principle why the author's GRD images derived from SLC (SLC>GRD) might be gridded at this resolution, but not those from GEE. Was some element of post-download image decimation/resampling carried out for consistency with the cell size of your SLC>GRD images? I suspect that more information must be added here for clarity.

L84. '*Specific format*'. What format? State here or reword sentence to remove this phrasing.

L90. In other words, TanDEM-X data is not available over Amery within DORIS at the time of processing. For concision, suggest rephrasing to more explicitly state that here.

Fig. 1 caption. Change '*Landsat 8 RGB images*' to '*Landsat 8 true colour images*' since RGB is ambiguous (could refer either to the actual red-green-blue bands or some combination of non-visible bands visualised across the RGB guns). This should be changed universally throughout the manuscript.

Fig 2. What does '*Write data in pixel interleaved 2b/2b complex short integer format*' mean? Clarify in the caption or rephrase to avoid jargon.

'*Compute the heights of the pixels in the radar coded system*'. Does this refer to some sort of unused phase unwrapping or conversion of phase into a DEM? If so, suggest removing here.

L96. Revise sentence for technical accuracy. Band 8 (panchromatic) has a native resolution of 15 m, not 30 m.

L102. For consistency with your SAR, optical and ERA5 data described elsewhere in the text, please state here the exact SSMIS product used and where you retrieved the data from.

Fig. 6. Fig.6 is only referenced after Figs. 7/8 in the text, so these figures should be reordered/numbered in the captions and text to reflect this.

L185. This is the first time that Fig. 6 is mentioned in the text, so I'm a little confused as to why panels a and b (melt over Amery) are shown. I presume this is an omission in the discussion above on Amery Ice Shelf, so the authors should add some brief discussion on melt as a potential cause (or, more likely, not) of the coherence change somewhere in the section between Lines 146 and 153.

**Technical Comments**

L4. Remove '*Therefore,*'.

L5. Remove '*C-band*' (this is later specified (and better placed) on L8.

L35. Should read: '*To the intuitive representation*'.

L41. Suggest rewriting sentence to read: '*InSAR processing uses pairs of images …*'.

L43. Remove '*considered*'.

L73-92. I appreciate the revisions made to this paragraph for scientific/methodological clarity, although suggest that the ordering of the paragraphs should be reversed to improve clarity (i.e. change discussion of GRD and then SLC to SLC>GRD). This is because, as clarified by the authors in the text, the GRD products "*are mainly used as supplementary backscatter intensity information when specific SLC tracks are not available*". In this regard, it seems peculiar to mention how data gaps were filled using GRD before an explicit discussion of the main data sources first (SLC).

Please also see my scientific comments pertaining to these paragraphs.

L78. '*The GRD data are primarily acquired …*'.

L80. Change '*recalled*' to '*termed sigma-nought ($\sigma^0$)*'.

L84. Change '*and is illustrated*' to '*whose processing chain is summarised in Fig. 2*'.

L88. Change '*is applied in addition*' to '*is also applied*' or similar.

Fig. 1 caption. Typo '*Fig. Figure 3*' and '*Fig. Figure 9*'. Check for this universally throughout the manuscript.

L129. Change '*levels*' to '*values*'.

L131. Change '*drop*' to '*decrease*'.

L150. Change '*from*' to '*of*'. Then, suggest revising this and next sentence to read: '*…with large temporal variations which fluctuate between 0.2 and 0.6 between successive (6-day) image acquisitions. These sudden drops likely result from short-term, weather-induced changes in scattering properties, including snowfall events (cf. Fig. 5a)*".

Fig. 3 caption. Does '*for all features presented*' mean '*from all features indicated in Fig 1*'? This isn't clear, so suggest revising to be more explicit.

Fig. 4 caption. Should read: '*over a quarter year of observations. The transects as well as the 2D winter appearance of the feature and its surroundings are illustrated in the bottom panels*'.

L154. Remove '*on the other hand, the coherence is lower as the*' as this is implicit from the following text. Also, please consider citing Table 1 somewhere in this sentence.

L156. Remove '*in a 12 day revisit*'. Suggest also changing '*Panel b) of Fig. 5*' to '*Fig. 5b*'. Also, suggest changing '*… stronger precipitation*' to '*… a greater amount of total daily precipitation*'.

L174. '*for only half*'.

L180. Suggest changing '*Differently from …*' to '*In contrast to data acquired over Amery ice Shelf ...*'.

Fig. 7 caption. Same comment regarding '*Landsat RGB images*' as in Fig. 1 caption.

Fig. 8. For consistency with Fig. 7, suggest possibly arranging this panel so that sigma-nought is on top, coherence in middle, and true colour Landsat 8 imagery on bottom (i.e. a 3rx2c layout as opposed to 2rx3c).

Fig. 8 caption. Suggest changing last sentence to '*Two near-contemporaneous Landsat true colour images are also shown*'. (See also my comments above on 'RGB' images).

L184. Please either refer to figure panels as '*Fig. X panel b)*' or '*Fig Xb*' and make this consistent throughout the manuscript.

L188. Change '*are sharpy emerging*' to '*sharply emerge*'.

L189. Change '*strips*' to '*curvilinear features*' or similar.

L190-192. For concision/clarity, suggest rephrasing these sentences to: '*This feature is not visible in either $\sigma^0$ or optical imagery, highlighting the benefit of InSAR-based coherence for the detection and monitoring of sub-surface lake networks*'.

L195. Change '*shows*' to '*is associated with*'.

L199. Suggest beginning sentence '*That both fringe …*'.

L206. I think '*forming a concentric pattern of deformation associated with a series of dense, closely spaced fringes*' would be better than '*a whirl-like feature*'.

L210. '*This would …*'. What would? Suggest rephrasing to: '*This hypothesis is consistent with earlier observations, including the rebound effects described by Banwell et al. (2013)*' to be more explicit.

L213-224. This paragraph is rather complex in that it tries to discuss backscatter, coherence and phase all at the same time and then, confusingly, introduces phase on line 222 before talking about the advantages of phase observations over backscatter/intensity.

For clarity, I would encourage the authors to first discuss the backscatter and how this compares to the coherence imagery, and then talk about phase (and its advantages in reducing ambiguities) afterwards. Restructuring the paragraph in this way would be more consistent with the broader layout of the results/discussion, which I think will help the readership.

L222. Incorrect grammar/sentence tense. Should read: '*Interferometric analyses show similar results (Fig. 10)*' (although see comment above – this sentence will likely be removed).

Fig. 9 caption. Try to avoid beginning sentences with mathematical symbols or other (uncommonly used) abbreviations. Should read: *'Sigma-nought ($\sigma^0$) and interferograms ...*". Same comment regarding Fig. 10.

L231. '… *based only on* …'.

L234. '…*may not therefore be…*'.

L238. '…*variations can indicate* …'.

L239. '…*may consequently be due to* …'.

L245. Could this be better worded for clarity, as I'm not sure exactly what this refers to? (to me, disc and rings are perfectly circular shapes …). The authors previously used *'polygonal features*' in is context, so suggest the same or similar term be used here for consistency.

L255. Suggest replacing '*revealed*' with '*imaged*' or '*constrained*' or similar.

L256. Suggest rewriting as: '… *is shown clearly in Fig. 9, where the closely spaced fringes shown could be used to estimate the presence of an uplift event due to drainage'*.

Note: I still think a rough estimate of uplift would be meaningful/interesting to the readership here, which you might also include in the appropriate section of the results. I accept the author's concerns that this would be a relative (LOS) displacement only, and that a lack of in-situ observations exist to validate this phenomenon, but as long as these caveats are also explicitly mentioned I think that this would be acceptable.

L260. '*e.g. strong snowfall events, as shown in Fig. 5)'*.

L264. Change '*done*' (colloquial) to '*performed*', '*carried out*' or similar.

L267. For clarity, suggest editing to read: '… *additional processing will likely be needed to cancel out, for example, the effects of ice-shelf flow (Mohajerani et al., 2021) ...*'.

L276. Grammar. Suggest rephrasing to: '*Four Antarctic ice shelf regions subject to intense summertime melt have been analysed using Sentinel-1A/B C-bad SAR imagery...*'.

L283. '*circumstances*'.

~END~

---

## Author Response (AR2)

Response to Referee 1 on tc-2021-169

We appreciate Referee 1 for additionally reviewing and commenting the manuscript. Please find the item-by-item reply below, with the original comments in *italics* and the responses in blue.

*I commend the authors for doing a fine job at addressing my concerns and for revising the manuscript accordingly. The manuscript is now much improved, and I believe that the addition of both the ERA5 precipitation and SSMIS passive microwave-derived melt estimates (added in response to some of Reviewer 2's concerns) are good supplements to the analyses presented. Below, I list several science and technical (mainly typos and grammatical) suggestions which should be incorporated into the final version of the manuscript. Line numbers refer to those in the revised manuscript. Once implemented, I believe the paper will be suitable for The Cryosphere and look forward to hopefully seeing it published in due course.*

We appreciate the comments. The typos and grammatical suggestions are implemented and noted as 'done' in this document.

**Scientific Comments**

*L45. Remove 'the whole or' (phase will never be a complete integer component of the wavelength'.*
(L44) Done.

*L53. 'Radiometric' isn't the correct word here since radiometry pertains to nearly all forms of remote sensing. Suggest 'optical and passive microwave satellite data' instead.*
(L57) Done.

*L71. I appreciate the authors response regarding this paragraph, although I'm afraid the final sentence still doesn't make much sense to me. If the Spergel et al. dataset exists, why didn't you use it? Perhaps it was not publicly available at the time of analysis? If so, this should be mentioned here, otherwise the sentence should be revised to improve clarity or removed.*
It is true that it was not publicly available at the time of analysis. However, to avoid confusion, this statement has been removed.

*L76. Following the author's responses to this sentence, I suggest stating explicitly here the fact that no data exist over Track 59 of Amery because this isn't available on the Copernicus Open Access Hub. (This isn't explicitly obvious from Table 1).*
Changed to 'data from ascending track 59 before July 2017 are not available on the Copernicus Open Access Hub, as shown in Table 1'.

*L79. Sentence beginning 'The final backscatter products'. Despite the author's responses to this sentence, I am still a little confused by this since the multi-looked cell size of the IW images on GEE are not 20x20 m (see https://developers.google.com/earth-engine/datasets/catalog/COPERNICUS_S1_GRD; "This collection contains all of the GRD scenes. Each scene has one of 3 resolutions (10, 25 or 40 meters"). I understand in principle why the author's GRD images derived from SLC (SLC>GRD) might be gridded at this resolution, but not those from GEE. Was some element of post-download image*

*decimation/resampling carried out for consistency with the cell size of your SLC>GRD images? I suspect that more information must be added here for clarity.*
According to the online documentation, the originally acquired resolution should be 10 m.

*L84. 'Specific format'. What format? State here or reword sentence to remove this phrasing.*
It is '.raw' format, according to the documentation. Since it is not an important step, this sentence has been removed.

*L90. In other words, TanDEM-X data is not available over Amery within DORIS at the time of processing. For concision, suggest rephrasing to more explicitly state that here.*
(L89) Changed to 'WGS84 geoid for Amery as it is the default DEM input of DORISwhen TanDEM-X DEM of the same quality is not available at the time of processing.'

*Fig. 1 caption. Change 'Landsat 8 RGB images' to 'Landsat 8 true colour images' since RGB is ambiguous (could refer either to the actual red-green-blue bands or some combination of non-visible bands visualised across the RGB guns). This should be changed universally throughout the manuscript.*
This has been changed to 'true colour' throughout.

*Fig 2. What does 'Write data in pixel interleaved 2b/2b complex short integer format' mean? Clarify in the caption or rephrase to avoid jargon.*
It is '.raw' format to be processed and saved by DORIS.
*'Compute the heights of the pixels in the radar coded system'. Does this refer to some sort of unused phase unwrapping or conversion of phase into a DEM? If so, suggest removing here.*
According to the online documentation: 'In this step in principle the heights in the radar coded system are computed. However with the exact method, the geocoding can be done in the same step.' This could be in the same step with geocoding, therefore is removed.

*L96. Revise sentence for technical accuracy. Band 8 (panchromatic) has a native resolution of 15 m, not 30 m.*
Different resolutions have been specified.

*L102. For consistency with your SAR, optical and ERA5 data described elsewhere in the text, please state here the exact SSMIS product used and where you retrieved the data from.*
Citation to National Snow and Ice Data Center (NSIDC) has been added.

*Fig. 6. Fig.6 is only referenced after Figs. 7/8 in the text, so these figures should be reordered/numbered in the captions and text to reflect this.*
Please refer to the next comment.

*L185. This is the first time that Fig. 6 is mentioned in the text, so I'm a little confused as to why panels a and b (melt over Amery) are shown. I presume this is an omission in the discussion above on Amery Ice Shelf, so the authors should add some brief discussion on melt as a potential cause (or, more likely, not) of the coherence change somewhere in the section between Lines 146 and 153.*
(L152) Added 'During summer, low coherence occurs when the surface melts. This can be seen in Fig. 6a and Fig. 6b, where XPGR exceeds the melting threshold in Jan. 2017, and rises

towards the melting threshold in Jan. 2018.' This is prior to the reference to Fig.7 and Fig. 8, therefore the order of the figures is kept.

***Technical Comments***
*L4. Remove 'Therefore,'.*
Done.

*L5. Remove 'C-band' (this is later specified (and better placed) on L8.*
Done.

*L35. Should read: 'To the intuitive representation'.*
Done.

*L41. Suggest rewriting sentence to read: 'InSAR processing uses pairs of images ...'.*
Done.

*L43. Remove 'considered'.*
Done.

*L73-92. I appreciate the revisions made to this paragraph for scientific/methodological clarity, although suggest that the ordering of the paragraphs should be reversed to improve clarity (i.e. change discussion of GRD and then SLC to SLC>GRD). This is because, as clarified by the authors in the text, the GRD products "are mainly used as supplementary backscatter intensity information when specific SLC tracks are not available". In this regard, it seems peculiar to mention how data gaps were filled using GRD before an explicit discussion of the main data sources first (SLC).*
*Please also see my scientific comments pertaining to these paragraphs.*
The orders have been reversed.

*L78. 'The GRD data are primarily acquired ...'.*
Done.

*L80. Change 'recalled' to 'termed sigma-nought ($\sigma$0)'.*
Done.

*L84. Change 'and is illustrated' to 'whose processing chain is summarised in Fig. 2'.*
Done.

*L88. Change 'is applied in addition' to 'is also applied' or similar.*
Done.

*Fig. 1 caption. Typo 'Fig. Figure 3' and 'Fig. Figure 9'. Check for this universally throughout the manuscript.*
Done.

*L129. Change 'levels' to 'values'.*
Done.

*L131. Change 'drop' to 'decrease'.*
Done.

*L150. Change 'from' to 'of'. Then, suggest revising this and next sentence to read: '…with large temporal variations which fluctuate between 0.2 and 0.6 between successive (6-day) image acquisitions. These sudden drops likely result from short-term, weather-induced changes in scattering properties, including snowfall events (cf. Fig. 5a)".*
Done.

*Fig. 3 caption. Does 'for all features presented' mean 'from all features indicated in Fig 1'? This isn't clear, so suggest revising to be more explicit.*
Changed to 'from all features indicated in Fig 1'.

*Fig. 4 caption. Should read: 'over a quarter year of observations. The transects as well as the 2D winter appearance of the feature and its surroundings are illustrated in the bottom panels'.*
Done.

*L154. Remove 'on the other hand, the coherence is lower as the' as this is implicit from the following text. Also, please consider citing Table 1 somewhere in this sentence.*
Changed to 'On RBIS, the Sentinel-1 data are only available in a 12-day revisit cycle (as in Table 1),…'

*L156. Remove 'in a 12 day revisit'. Suggest also changing 'Panel b) of Fig. 5' to 'Fig. 5b'. Also, suggest changing '… stronger precipitation' to '… a greater amount of total daily precipitation'.*
Done.

*L174. 'for only half'.*
Done.

*L180. Suggest changing 'Differently from …' to 'In contrast to data acquired over Amery ice Shelf …'.*
Done.

*Fig. 7 caption. Same comment regarding 'Landsat RGB images' as in Fig. 1 caption.*
Done.

*Fig. 8. For consistency with Fig. 7, suggest possibly arranging this panel so that sigma-nought is on top, coherence in middle, and true colour Landsat 8 imagery on bottom (i.e. a 3rx2c layout as opposed to 2rx3c).*
Done. The caption is also changed accordingly.

*Fig. 8 caption. Suggest changing last sentence to 'Two near-contemporaneous Landsat true colour images are also shown'. (See also my comments above on 'RGB' images).*
Done.

*L184. Please either refer to figure panels as 'Fig. X panel b)' or 'Fig Xb' and make this consistent throughout the manuscript.*
Done.

*L188. Change 'are sharpy emerging' to 'sharply emerge'.*
Done.

*L189. Change 'strips' to 'curvilinear features' or similar.*
Done.

*L190-192. For concision/clarity, suggest rephrasing these sentences to: 'This feature is not visible in either $\sigma 0$ or optical imagery, highlighting the benefit of InSAR-based coherence for the detection and monitoring of sub-surface lake networks'.*
Done.

*L195. Change 'shows' to 'is associated with'.*
Done.

*L199. Suggest beginning sentence 'That both fringe ...'.*
Done.

*L206. I think 'forming a concentric pattern of deformation associated with a series of dense, closely spaced fringes' would be better than 'a whirl-like feature'.*
Done.

*L210. 'This would ...'. What would? Suggest rephrasing to: 'This hypothesis is consistent with earlier observations, including the rebound effects described by Banwell et al. (2013)' to be more explicit.*
Done.

*L213-224. This paragraph is rather complex in that it tries to discuss backscatter, coherence and phase all at the same time and then, confusingly, introduces phase on line 222 before talking about the advantages of phase observations over backscatter/intensity.*
*For clarity, I would encourage the authors to first discuss the backscatter and how this compares to the coherence imagery, and then talk about phase (and its advantages in reducing ambiguities) afterwards. Restructuring the paragraph in this way would be more consistent with the broader layout of the results/discussion, which I think will help the readership.*
The whole paragraph has been re-written.

*L222. Incorrect grammar/sentence tense. Should read: 'Interferometric analyses show similar results (Fig. 10)' (although see comment above – this sentence will likely be removed).*
This sentence has been removed.

*Fig. 9 caption. Try to avoid beginning sentences with mathematical symbols or other (uncommonly used) abbreviations. Should read: 'Sigma-nought ($\sigma 0$) and interferograms ...".*
*Same comment regarding Fig. 10.*

Done.

*L231. '... based only on ...'.*
(L238) Done.

*L234. '...may not therefore be...'.*
(L241) Done.

*L238. '...variations can indicate ...'.*
(L245) Done.

*L239. '...may consequently be due to ...'.*
(L246) Done.

*L245. Could this be better worded for clarity, as I'm not sure exactly what this refers to? (to me, disc and rings are perfectly circular shapes ...). The authors previously used 'polygonal features' in is context, so suggest the same or similar term be used here for consistency.*
(L252) The disc shape refers to the whole lake, while the ring shape refers to only the edge. This sentence has been changed to 'The change from complete polygonal low coherence patterns to partly high coherence…'

*L255. Suggest replacing 'revealed' with 'imaged' or 'constrained' or similar.*
(L263) Replaced with 'imaged'.

*L256. Suggest rewriting as: '... is shown clearly in Fig. 9, where the closely spaced fringes shown could be used to estimate the presence of an uplift event due to drainage'.*
*Note: I still think a rough estimate of uplift would be meaningful/interesting to the readership here, which you might also include in the appropriate section of the results. I accept the author's concerns that this would be a relative (LOS) displacement only, and that a lack of in-situ observations exist to validate this phenomenon, but as long as these caveats are also explicitly mentioned I think that this would be acceptable.*
Done. Added 'By counting the fringes, the feature consists of approximately 7 fringes, each measuring 2.8 cm in the line-of-sight. Assuming a vertical movement, this corresponds to an uplift of approximately 24 cm(taking into account an incidence angle of approximately 35°). However, this amount of uplift is only an approximation of a displacement relative to ice flow and tidal component, and needs in situ observations to validate.' to L213.

*L260. 'e.g. strong snowfall events, as shown in Fig. 5)'.*
(L269) Done.

*L264. Change 'done' (colloquial) to 'performed', 'carried out' or similar.*
(L273) Changed to 'performed'.

*L267. For clarity, suggest editing to read: '... additional processing will likely be needed to cancel out, for example, the effects of ice-shelf flow (Mohajerani et al., 2021) ...'.*
(L276) Done.

*L276. Grammar. Suggest rephrasing to: 'Four Antarctic ice shelf regions subject to intense summertime melt have been analysed using Sentinel-1A/B C-bad SAR imagery...'.*
(L285) Done.

*L283. 'circumstances'.*
(L292) Done.

Response to Editor
We also would like to appreciate Dr. Chris Derksen for editing this manuscript.